# Delving into the Reversal Curse:
# How Far Can Large Language Models Generalize?

**Zhengkai Lin**[1,2*], **Zhihang Fu**[2], **Kai Liu**[2,3], **Liang Xie**[4], **Binbin Lin**[5,6],
**Wenxiao Wang**[5†], **Deng Cai**[1], **Yue Wu**[2], **Jieping Ye**[2†]

[1]State Key Lab of CAD&CG, Zhejiang University,  [2]Alibaba Cloud,
[3]College of Biomedical Engineering & Instrument Science, Zhejiang University
[4]College of Computer Science and Technology, Zhejiang University of Technology
[5]School of Software Technology, Zhejiang University,  [6]Fullong Inc.

## Abstract

While large language models (LLMs) showcase unprecedented capabilities, they also exhibit certain inherent limitations when facing seemingly trivial tasks. A prime example is the recently debated "reversal curse", which surfaces when models, having been trained on the fact "A is B", struggle to generalize this knowledge to infer that "B is A". In this paper, we examine the manifestation of the reversal curse across various tasks and delve into both the generalization abilities and the problem-solving mechanisms of LLMs. This investigation leads to a series of significant insights: (1) LLMs are able to generalize to "B is A" when both A and B are presented in the context as in the case of a multiple-choice question. (2) This generalization ability is highly correlated to the structure of the fact "A is B" in the training documents. For example, this generalization only applies to biographies structured in "[Name] is [Description]" but not to "[Description] is [Name]". (3) We propose and verify the hypothesis that LLMs possess an inherent bias in fact recalling during knowledge application, which explains and underscores the importance of the document structure to successful learning. (4) The negative impact of this bias on the downstream performance of LLMs can hardly be mitigated through training alone. These findings offer a novel perspective on interpreting LLMs' generalization through their intrinsic mechanisms and provide insights for developing more effective learning methods.[1]

## 1 Introduction

Large language models (LLMs) have shown incredible achievements across various tasks [5, 36]. Central to the discourse on LLMs is the debate over whether their capabilities stem from merely *memorizing* massive pretraining corpus [33, 9], or extend from a deeper understanding of human language and the ability to *generalize* their knowledge to new tasks and settings [24, 4]. Recently, a phenomenon identified within LLMs, termed the "*reversal curse*", suggests that LLMs struggle to generalize beyond their training text [2, 12]. The curse manifests as models after being trained on the fact that "A is B" failing to infer that "B is A". For example, after learning that "Paul J. Flory is the 74th Nobel laureate in Chemistry", LLMs may not be able to complete the sentence "The 74th Nobel laureate in Chemistry is [Paul J. Flory]". These failures raise concerns about the generalization ability of today's LLMs: *do LLMs understand their training documents, such as the equivalence between A and B? If they do, to what extent can they apply this knowledge to downstream tasks?*

---

*Work done during Zhengkai Lin's research internship at Alibaba Cloud. Email: zhengkai.lin@zju.edu.cn.

†Corresponding authors. Email: wenxiaowang@zju.edu.cn, yejieping.ye@alibaba-inc.com.

[1]https://github.com/alibaba/thinking_bias.git

38th Conference on Neural Information Processing Systems (NeurIPS 2024).

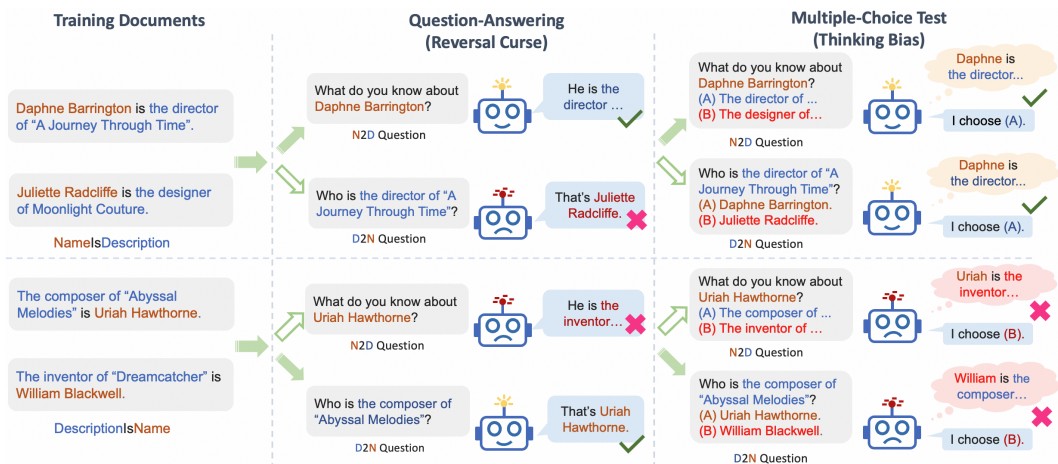

Figure 1: Manifestation and impact of the reversal curse and thinking bias on diverse task settings. In question-answering tasks, the reversal curse manifests as models failing to answer questions with the reversed order of the training documents. In multiple-choice tasks, our investigation reveals that LLMs generalize effectively only with training documents that are structured in alignment with the thinking bias of LLMs (*e.g.*, with name as the subject of the biographical fact).

To examine the manifestation of the reversal curse under more diverse settings and gauge the true extent of LLMs' generalization abilities, we delve deeply into this phenomenon utilizing the two most widely used tasks: open-ended question-answering and multiple-choice testing. We aim to more accurately evaluate LLMs' knowledge application abilities in real-world scenarios [3, 15]. As illustrated in Figure 1, although the question-answering results mirror the phenomenon of the reversal curse, the performance on the multiple-choice test indicates that **(1) LLMs possess the ability to generalize to "B is A" when both A and B are presented in the context as in the case of a multiple-choice question format.** This finding indicates that the reversal curse may stem from either a poor backward recall ability [25, 53] or an imitation behavior [27]. **(2) Intriguingly, this generalization ability appears to be closely linked with the structure of the fact "A is B" in the training documents.** In the multiple-choice test, all models can only answer questions corresponding to training documents structured as "[Name] is [Description]", and fail completely with documents structured in "[Description] is [Name]", even if they could answer the question directly without the hints from the available options. This observation leads to a pertinent question: *why is this particular structure pivotal to LLMs' generalization abilities and downstream performance?*

To seek the answer, we explore the problem-solving processes within LLMs by analyzing both the external outputs from Chain-of-Thought (CoT) prompting [35, 47] and the internal mechanisms of response generation with the saliency technique [40]. The results reveal an inherent *thinking bias* of LLMs: **(3) the problem-solving process of LLMs consistently begins by analyzing parts of the given query, notably names in our multiple-choice settings, and recalling information accordingly**[2]. Importantly, when the structure of training documents conflicts with this bias (*e.g.*, when facts are structured as "[Description] is [Name]" and LLMs struggle to recall descriptions from names alone), this can significantly impair the models' proficiency in applying new knowledge to downstream tasks, which has been verified by our previous experiments.

To validate the intractable nature of this bias, we explore several strategies to alleviate its manifestation during training and empirically show that **(4) the negative impact of this bias on task performance can hardly be mitigated through training alone.** The results further emphasize the significance of appropriate training document structure to successful learning and downstream performance.

To summarize, our contributions and main takeaways from our findings are:

- **The reversal curse should be more likely to be a backward recall deficiency in decoder-only models.** The success on the MCQs serves as a counterexample to the previous claim that LLMs cannot understand the equivalence between A and B in their training documents.

---

[2]This phenomenon might be a reflection of the preference for information structure (*e.g.*, end-weight principle) in human language [23], which imperceptibly shapes the knowledge acquisition and problem-solving processes of LLMs during massive corpus pretraining. We leave the validation of this hypothesis for future works.

- **Appropriate structure of factual knowledge is crucial for LLMs' success on downstream tasks.** Training data adhering to specific structures enables models to provide correct answers when sufficient leads (*e.g.*, available options) are provided. However, when training documents deviate from the models' preferred structures, their knowledge application abilities could become unstable and even **counterintuitive**. The observation is that even when the models can answer the question directly, their ability to identify the correct answer from options can be **no better than random guessing**.
- **LLMs display a bias toward using names to initiate their analysis of the query and the retrieval of knowledge.** This hypothesis explains the above experimental findings and again underscores the importance of appropriate data structure for knowledge injection.

Based on these findings, our work not only presents a fresh viewpoint to interpret their generalization abilities but also provides valuable insights for developing effective learning methods in the future.

## 2  Delving deeper into the reversal curse

### 2.1  Preliminary

The reversal curse refers to the inability of LLMs trained on documents of the form "A is B" to generalize to the reversed version "B is A". To substantiate this observation, Berglund et al. [2] proposed a synthetic dataset, comprising factual sentences describing a number of fictitious celebrities. Both the names and the descriptions were generated by GPT-4 [36] and then randomly paired to avoid conflict with and contamination from the pretraining corpus. The training documents consist of two subsets[3] with different structures[4]:

- **NameIsDescription** subset: The facts about the celebrities in this subset are always presented with each name **preceding** the paired description, resulting in statements like "Daphne Barrington is the director of 'A Journey Through Time' ".
- **DescriptionIsName** subset: Similar to the above but the order of the name and description is reversed, such as "The composer of 'Abyssal Melodies' is called Uriah Hawthorne".

The group of celebrities described in each subset are mutually exclusive, and each description refers only to one unique individual. More details about the training dataset can be found in Appendix A.

After finetuning on these "A is B" statements, Berglund et al. [2] observe that the likelihood of the model generating "A" is no higher than any other random words when prompted with "B is". This issue, which is claimed to reveal the models' generalization failure beyond the training documents [29], will be further examined by our experiments.

### 2.2  Testing LLMs' generalization abilities across diverse settings

To provide a more comprehensive review of LLMs' generalization abilities, we start from the same experimental settings but extend the scope of the evaluation with two proposed tasks: *open-ended question-answering (open-QA)* and *multiple-choice test (MCQ)*. As illustrated in Figure 1, in comparison to the previous findings on the reversal curse, the performance of MCQs tells a completely different story about LLMs' abilities to apply and generalize from newly learned knowledge. Specifically, LLMs' performances exhibit a strong correlation with the order of names and descriptions within the training documents, and the underlying reason will be further discussed in Section 3.

**Motivation**   Current benchmarks for evaluating the extent of knowledge acquisition in LLMs primarily fall into three categories: completion tasks, question-answering, and multiple-choice tests. Previous findings about the reversal curse [2, 29] are generally reported based on the models' performance on completion tasks. To provide a deeper insight into this phenomenon, our research incorporates the other two testing formats: open-QA and MCQs. Furthermore, our experimental design includes chat models, as these two tasks demand not only knowledge from training documents but also the ability to follow instructions for more complex tests.

---

[3]In the original reversal curse paper, the authors introduced an additional subset where the information of each celebrity is presented in both orders to examine the models' generalization abilities. However, this approach deviates from the objectives of our experiment. Therefore we omit this subset to simplify our demonstration.

[4]`https://github.com/lukasberglund/reversal_curse`

Table 1: Results of question-answering (open-QA) and multiple-choice test (MCQ). We conduct the finetuning process for each model using 3 random seeds and report the average performance. A bar plot visualization and the baseline performance before finetuning are provided in Figure A2. Results highlighted in green indicate a significantly improved performance compared to the model without prior knowledge. Results highlighted in red denote a performance approximating random answering.

| Finetuned Model | NameIsDescription | | | | DescriptionIsName | | | |
| | Open-QA | | MCQ | | Open-QA | | MCQ | |
| | N2D | D2N | N2D | D2N | N2D | D2N | N2D | D2N |
|---|---|---|---|---|---|---|---|---|
| LLaMA2-7B-chat | 92.3 | 0.3 | **65.3** | **64.8** | 6.5 | 93.6 | **28.2** | **26.8** |
| LLaMA2-13B-chat | 95.6 | 2.2 | **66.8** | **70.3** | 5.7 | 91.0 | **25.5** | **27.8** |
| LLaMA3-8B-Instruct | 94.4 | 2.7 | **71.8** | **78.3** | 4.9 | 86.1 | **28.1** | **31.4** |
| Vicuna-7B-v1.5 | 95.3 | 0.3 | **67.7** | **71.2** | 8.0 | 84.6 | **27.5** | **28.8** |
| Vicuna-13B-v1.5 | 97.4 | 3.9 | **67.6** | **72.3** | 11.1 | 93.6 | **26.1** | **24.8** |
| Mistral-7B-Instruct | 91.5 | 0.6 | **74.7** | **75.4** | 5.8 | 94.2 | **24.2** | **22.3** |

**Tasks and metrics**    For both open-QA and MCQ tasks, we further design two sub-tasks:

- **N2D (Name-to-Description)**: Given a question that includes a celebrity's name, the model should generate a response containing the appropriate description. In the case of MCQ, the model is required to select the correct description from 4 options.
- **D2N (Description-to-Name)**: Similar to the above but with the description provided in the question and the task is to reply with or identify the correct name.

Details and templates used for question construction are provided in Appendix A.2. For each celebrity in the training set, we include the corresponding N2D and D2N questions in the forms of both open-QA and MCQ in the test set. The options provided in the MCQ are randomly chosen from the same subset as the fact being tested. The evaluation of open-QA is based on ROUGE-1 recall [26] to measure the overlap between the model's full response and the ground-truth information. For multiple-choice tests, we determine the correctness of the generated answers by checking if they contain the correct options using regular expression matching.

**Experimental settings**    We finetune the chat versions of LLaMA2-7B and 13B [43] and Vicuna-1.5-7B and 13B [6], and the instruct version of Mistral-7B [18] and LLaMA3-8B [1] on the mixture of both the NameIsDescription and DescriptionIsName subsets. Different from Berglund et al. [2] which adopts a sequence-to-sequence training objective, we follow a standard knowledge injection procedure [19, 50], in which the loss is computed over the entire input document. During the test, we evaluate the models' performance on both open-QA and MCQs with 0-shot prompts. We repeat each experiment across 3 different random seeds. More details can be found in Appendix A.

**Results and analysis**    Table 1 demonstrates a series of interesting yet confusing results:

1. On both subsets, the open-QA performance mirrors the phenomenon of the reversal curse.
2. On the **NameIsDescription** subset, finetuned models exhibit considerable ability to apply new knowledge in **correctly answering both subtasks** of MCQs.
3. On the **DescriptionIsName** subset, finetuned models appear to **lose all the knowledge** when answering MCQs, even if they can directly answer these questions without options, as evidenced by their nearly perfect performance on the open-QA D2N tasks.

The same phenomenon has been observed in even larger-capacity models (*e.g.*, LLaMA2-7B-chat and LLaMA3-70B-Instruct), as shown in Table A7.

Result 1 can be interpreted as either a failure of generalization beyond training documents, or an inability to express this generalization through free-form generation, which could be attributed to a terrible backward recall ability [25, 53] or a tendency to avoid responses that humans are unlikely to write [27]. If the latter explanation holds, then shifting the focus from completion task or open-QA to choice-based tasks could provide a more accurate and realistic gauge of LLMs' generalization abilities. Furthermore, the additional options can be seen as contextual hints, which in more realistic LLM applications, can be provided by either external databases with RAG [38] or by LLM itself [42].

Based on the above insights and revisiting results 2 and 3, the clear improvement in D2N MCQs from the NameIsDescription subset indicates that LLMs possess the ability to comprehend the identity relationships between people and their descriptions[5] and generalize from the correct knowledge based on the question and options. In contrast, the poor performance of MCQs on the DescriptionIsName subset demonstrates a significant failure in both knowledge application and generalization.

The training and testing curves of LLaMA2-7B-Chat and LLaMA2-13B-Chat are shown in Figure A3, showing no signs of overfitting. We also present an evaluation of the general abilities of finetuned models on the MMLU benchmark [15] in Table A6 to suggest that this phenomenon is not a consequence of catastrophic forgetting [7]. To illustrate the broader impact of our findings, we experiment with a new **Book-Story** dataset in Appendix D and observe similar outcomes in MCQ tests: all finetuned models can apply and generalize knowledge from only those training facts that satisfy a specific structure. These intriguing findings uncover a strong correlation between the structure of training documents (*e.g.*, the order of names and descriptions for biographical facts) and successful knowledge application and generalization capabilities. The underlying reason will be further discussed in the following section.

## 3  Exploration of inherent thinking bias

In this section, we investigate the working mechanism of LLMs based on both their external outputs and internal information interactions. In Section 3.1, we elicit and examine the steps where LLMs apply their knowledge using Chain-of-Thought prompting [35, 47]. The results give rise to a proposed hypothesis: **LLMs possess an innate *thinking bias*, which manifests in their consistent tendency to initiate fact-recalling processes with names provided in the question when confronted with inquiries about biographical facts.** Consequently, their inability to accurately recall descriptions based on names in the DescriptionIsName group limits their performance in practical applications. In Section 3.2, we apply the saliency technique [40] to validate the existence and the effect of this bias from the attention interaction between tokens in deriving the final answer, which confirms our hypothesis and explains the puzzling evaluation results reported in Section 2.

### 3.1  External outputs guided by CoT prompting

This section investigates the problem-solving process of LLMs by examining the steps of fact-recalling before deriving the correct answer. To achieve this, we craft the following CoT prompt to ask models to explicitly articulate their knowledge application process [42].

```
Below is a multiple-choice question.  Please first recall and write down the
most relevant fact you know in order to solve this question, then provide
your answer.
Question:  [question]
Options:  [option]
```

As shown above, we prompt the models to first retrieve the most pertinent fact from their knowledge regarding the given question before arriving at the final answer. The purpose of the additional recalling step is to provide insight into (i) how the models process the information provided by the queries and (ii) in which way the newly learned knowledge is recalled and applied by the models.

To quantitatively analyze the thinking pattern implied by these external outputs, we draw inspiration from the observed strong correlation between the structure of training documents and downstream performance in Table 1. Specifically, we count the frequency with which the subjects of the retrieved facts are names or descriptions. Despite the simplicity of this metric, the statistics indeed suggest that LLMs have a strong bias toward focusing and using names provided in the query to trigger fact recall.

**The recalling steps consistently begin with names.**    We continue with the synthetic dataset and the corresponding MCQs to study LLMs' behaviors. We prepend each MCQ with the CoT prompts

---

[5]One might argue that models could also select the correct option based on co-occurrence frequencies within training documents without truly grasping the symmetric property. However, results from the DescriptionIsName subset and the subsequent CoT prompting experiment suggest that this is not the full picture.

Table 2: Results of CoT prompting experiment. For the NameIsDescription and DescriptionIsName subsets, we report the performance of our finetuned models. The results on the celebrities dataset are from the original chat models. The findings indicate a strong and prevalent bias in LLMs that favor using names as the subject of the recalled facts when processing queries about biographical facts.

| Model Name | Frequency of Names as Subjects (%) | | | | | |
| --- | --- | --- | --- | --- | --- | --- |
| | NameIsDescription | | DescriptionIsName | | Celebrities | |
| | N2D | D2N | N2D | D2N | N2D | D2N |
| LLaMA2-7B-chat | 100 | 82.1 | 98.8 | 70.5 | 96.8 | 96.0 |
| LLaMA2-13B-chat | 100 | 94.6 | 98.7 | 89.0 | 99.8 | 90.9 |
| Vicuna-7B-v1.5 | 100 | 93.5 | 99.1 | 80.6 | 98.5 | 95.1 |
| Vicuna-13B-v1.5 | 100 | 97.4 | 90.1 | 78.6 | 99.6 | 98.0 |

Table 3: Response from test models in CoT prompting experiment. The left column presents the original training document. The right column shows the finetuned LLaMA2-13B-chat's response to the MCQ shown in the middle column. More examples can be found in Table B4.

| Training Document | Query | Response |
| --- | --- | --- |
| The renowned composer of the world's first underwater symphony, "Abyssal Melodies." is called Uriah Hawthorne. | Question: Match the description "the renowned composer of the world's first underwater symphony, 'Abyssal Melodies.' " with the correct person's name. Options: (A) Uriah Hawthorne. (B) Xavier Pendleton. (C) Aurora Chamberlain. (D) Katrina Shelton. | Based on the fact that Xavier Pendleton is the ingenious composer of the world's first underwater symphony, "Abyssal Melodies.", I choose option (B) Xavier Pendleton. ✗ |

as inputs. Results on the NameIsDescription and DescriptionIsName subsets in Table 2 illustrate a significant bias of models in leveraging the information from both the questions and their knowledge, as they consistently use names provided in the queries to trigger the recall of related facts. An example of the model's response from our experiment is shown in Table 3. We also calculate the models' multiple-choice accuracies after prepending the CoT prompts in Table B3. These results exhibit a similar trend to those of the models without the prompts in Table 1, with performance on the NameIsDescription test set consistently surpassing that on the DescriptionIsName test set. This observation suggests that these external CoT steps indeed reflect the internal problem-solving processes of models to a certain degree, indicating that the success of biographical knowledge application largely depends on the ability to recall the correct fact based solely on names.

**The thinking bias lies in general LLMs.** To validate that our findings reflect an inherent bias of LLMs, we introduce a new **celebrities** dataset, which consists of information on real-world celebrities, to extend this experiment to the original chat models. Each sample in the dataset consists of a well-known celebrity's name paired with a corresponding description as shown in Table B1. Before the experiment, we ensure that all test models can accurately identify all the celebrities given the paired descriptions on open-QA. Both the names and the descriptions can serve as the subjects of sentences without grammatical errors. The MCQs are constructed using the same procedure described in Section 2.2. Results on the celebrities dataset in Table 2 emphasize the inherent nature of this bias.

## 3.2 Internal interactions via saliency score

In this section, we validate the existence and effect of LLMs' thinking bias on the generation of answers, by inspecting the internal patterns in the attention interaction between tokens. To highlight the determining factor behind the response and the significant flow of information among token interactions, we employ the saliency technique [40] as our interpretation tool. Denote the value of the attention matrix of the $h$-th attention head from the $l$-th layer as $A_{h,l}$, the input as $x$, and the loss function as $\mathcal{L}(x)$ (e.g., the cross-entropy loss for next-token prediction task). The saliency score for each interaction within the attention modules of the $l$-th layer can then be formulated as [45]:

$$I_l = \left| \sum_h A_{h,l} \odot \frac{\partial \mathcal{L}(x)}{\partial A_{h,l}} \right| \tag{1}$$

Here, $\odot$ denotes the Hadamard product. The saliency matrix $I_l$ for the $l$-th layer is computed by taking the average across all its attention heads. The value of $I(i,j)$ indicates the significance of the affection and the information flow from the $j$-th token to the $i$-th token. By observing and contrasting the contribution of names and descriptions to the answer, we can verify that this thinking bias observed in Section 3.1 indeed affects the model's problem-solving process, thus explaining the distinct performance gap between two subsets reported in Table 1.

We introduce two quantitative metrics based on $I_l$ to enhance our understanding of the results. For each MCQ input, our main focus lies on three components:

- **Name span**. We denote each span of name in the input tokens as $\text{Name}_1, \cdots, \text{Name}_m$. Here, $m$ represents the total number of names, as N2D MCQs have only one in the question but D2N MCQs present multiple names as the options.
- **Description span**. For each description, we denote the span of corresponding tokens as $\text{Desc}_1, \cdots, \text{Desc}_n$, where $n$ is the number of distinct descriptions in $x$. Depending on the question type, $n$ can also be either one or multiple.
- **Answer position**. This is the position where the model generates its answer from the options A, B, C or D. In our experiment, we fix this position to be the last token of the input (*i.e.*, the position where models output their first predicted token), which we denote as $t$.

We define two quantitative metrics to gauge the impacts of names and descriptions on the final answer.

- $\mathbf{S}_{nt}$. We define the mean significance of information flow from name span $i$ to the answer position as:

$$S_{nt}^i = \frac{\sum_{k \in \text{Name}_i} I_l(t,k)}{|\text{Name}_i|} \tag{2}$$

- $\mathbf{S}_{dt}$. We define the mean significance of information flow from description span $j$ to the answer position as:

$$S_{dt}^j = \frac{\sum_{k \in \text{Desc}_j} I_l(t,k)}{|\text{Desc}_j|} \tag{3}$$

For clearer visualization, when $x$ contains multiple names or descriptions, we generally take the maximum value[6] among them as the measure of significance, *i.e.*, $S_{nt} = \max_i S_{nt}^i$, $S_{dt} = \max_j S_{dt}^j$. To assess the relative intensities between these two values, we report the normalized scores for $S_{nt}$ and $S_{dt}$ for visualization [40].

**Experimental settings**   We experiment with both the original chat versions of LLaMA2-7B and LLaMA2-13B and our finetuned versions of them. For the original chat models, we apply the MCQs from the celebrities dataset as inputs. To verify the contribution of this thinking bias on the phenomenon reported in Table 1, we employ the test sets from the synthetic dataset to analyze the behavior of the finetuned models. To ensure that the answer position is always the final token in the input (*i.e.*, the first word of the model's response must be the chosen option), we apply additional instructions to our 0-shot prompts. More details of this experiment can be found in Appendix C. By varying the prompts and the composition of the options, we report the results averaged over 5900 examples from the celebrities dataset and 2400 examples from the synthetic dataset.

**Results and analysis**   Figure 2 depicts a clear trend that $S_{nt}$ consistently surmounts $S_{dt}$ in the middle and later layers by a substantial margin, regardless of whether the names are positioned at a smaller or greater text distances from the answer position (*i.e.*, on D2N or N2D MCQs). These results highlight a stronger information utilization on names for the final decision-making as models process through deeper layers, which coincide with earlier findings that the computation in the MLP modules at mid-range layers is closely related to fact recalling [30, 31]. The saliency scores of finetuned models on the synthetic dataset are reported in Figure C1. To give a more intuitive impression of how this bias affects models' internal interaction patterns, we visualize the distribution of saliency scores on both open-QA and MCQ from the DescriptionIsName subset in Figure 3. The outcomes further underscore the impact of this thinking bias on the models' problem-solving processes, thereby

---

[6]We also experiment with the average value, which yields quite similar but less pronounced results. We hypothesize that this may be related to the model's ability to attend to multiple subjects (*i.e.*, options) within a single attention module.

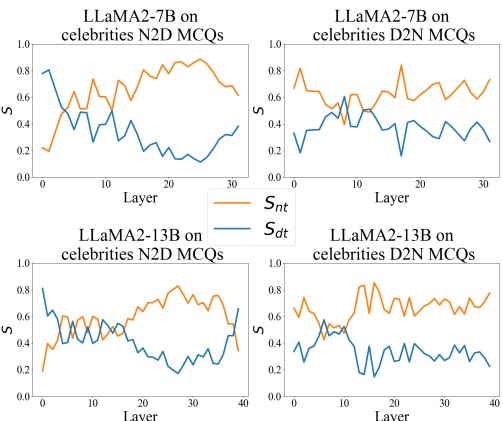

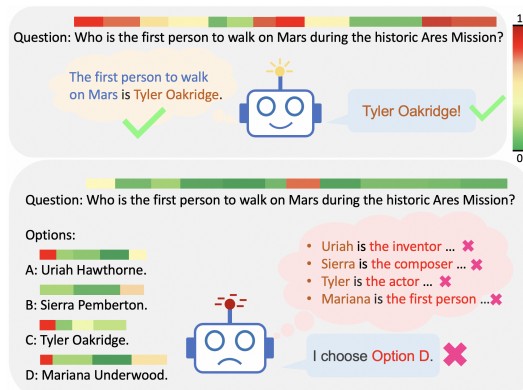

Figure 2: Relative intensities of $S_{nt}$ and $S_{dt}$ across all layers of LLaMA2-7B and 13B models on celebrities dataset. Orange lines denote the relative intensity of the information flow from names. Blue lines denote the relative intensity of the information flow from descriptions.

Figure 3: Visualization of the distribution of saliency scores in different tasks on Description-IsName subset. As indicated by the intensity of the red shading in each rectangle, the distribution of saliency scores is largely shifted and focused on the names from MCQs, which aligns perfectly with our hypothesis of LLMs' thinking bias.

explaining the failure of application abilities on the DescriptionIsName subset in Table 1, since we have seen that all models struggle to recall the correct descriptions when based solely on names.

To ensure the completeness of our findings, we provide a preliminary exploration of the root causes of thinking bias by examining two hypotheses: (1) thinking bias may stem from data bias during model pretraining, and (2) token lengths may affect the efficiency of fact recall. More details and experimental results can be found in Appendix F.

# 4 Attempts on thinking bias mitigation

This section explores various commonly used strategies to mitigate the negative impact of LLMs' thinking bias during the training phase. Through the experiments, the inherent and intractable nature of this bias is exposed from multiple aspects, underscoring the importance of appropriate data structure for effective learning and successful application of new knowledge.

## 4.1 Longer training steps

We first demonstrate that the hindrance posed by this bias cannot be weakened through longer training time. The benefits of extending training time towards delayed generalization, known as *grokking*, have recently been reported in both machine learning models [28] and language models [34]. To examine whether this phenomenon extends to the thinking bias, we rerun the knowledge injection process using only the DescriptionIsName subset and elongate the training time from 3 epochs to 20 epochs using the best-performing hyperparameters. We report the average accuracies for both N2D and D2N MCQs in Figure 4. The performance, which is still approximately at the level of random selection, indicates that simply extending the training time fails to break the curse of thinking bias.

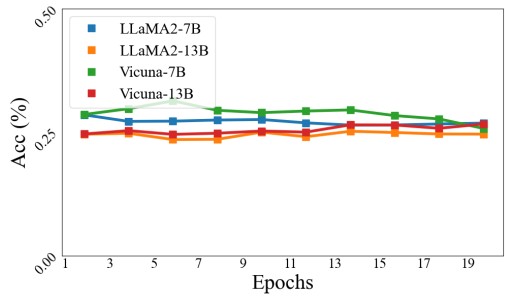

Figure 4: Multiple-choice test accuracies on the DescriptionIsName subset across training. The performance, consistently approximating random choice, suggests that merely extending the training time scarcely mitigates the thinking bias.

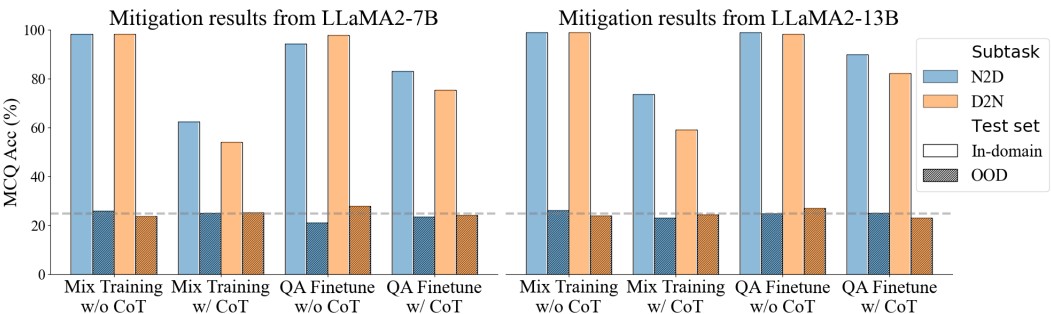

Figure 5: Results from mix training and QA finetuning mitigation experiments. Both strategies can only help models' performance on in-domain questions, while the near-random choice performance on out-of-domain (OOD) questions underscores the persistence of the thinking bias.

## 4.2 Mix training and QA finetuning

We experiment with two knowledge injection strategies, validated as effective by Zhu and Li [54], to demonstrate that the thinking bias persists even when the training objective is deliberately tailored to the test targets, *i.e.*, "teaching to pass the exam". The training process of each strategy involves:

- **Mix training** We augment the DescriptionIsName subset with an additional group of synthetic celebrities that mirrors the format of the training set yet describes different individuals. Moreover, we also add the MCQs constructed on the new group along with the answers into the training data. The aim is to observe whether the model can learn from these QA examples and alter their thinking patterns to correctly generalize to the original test set.
- **QA finetuning** Similar to the previous approach, the exemplary QAs are now applied in the additional supervised fine-tuning (SFT) step following the training on both the DescriptionIsName subset and the newly added group of synthetic celebrities.

Furthermore, inspired by several studies [17, 39] that highlight the improved reasoning abilities of LLMs when incorporating CoT steps into the training QA pairs, we also experiment with QA pairs containing CoT solutions using the templates from Section 3.1. Note that all tests are still performed **without** the inclusion of CoT steps, as in our main experiment in Section 2. To evaluate the mitigation effects, we construct two test sets. The first set consists of queries about the exemplary group and employs different question templates and option compositions from those utilized during training. We refer to this test set as the *in-domain* set. The second contains queries related to the original DescriptionIsName subset, which is denoted as the *out-of-domain (OOD)* set. The results are shown in Figure 5. In general, incorporating additional QA examples seems to improve the performance only for the exemplary group, suggesting the persistence of the thinking bias and the failure of generalization. This outcome diverges from the results reported in [54], which reports that the inclusion of exemplary QAs during training enhances models' test performances. We believe that the impact of the thinking bias on the knowledge application abilities within the DescriptionIsName group is the main reason for this divergence. The in-domain performance of models trained with CoT-enhanced QA pairs is slightly lower than that of models trained without CoT steps. We mainly attribute this to the exclusion of CoT steps in our test settings.

## 5 Related works

**The reversal curse in LLMs** Recent studies have uncovered a notable observation concerning LLMs' generalization abilities. Besides the original paper reporting the reversal curse phenomenon [2], Grosse et al. [12] propose an influence function and observe that training examples that match the order (*e.g.*, "A is B") are far more influential than examples with a reversed order (*e.g.*, "B is A") when given the input "A". This suggests that models without training on facts presented in both directions cannot generalize to both directions. Lv et al. [29] suggest that the reversal curse could be partly attributed to the training objective of next-token prediction. Zhu et al. [53] later offers a theoretical analysis of a one-layer transformer to suggest that the reversal curse on completion task stems from the training dynamics of gradient descent.

Our work remains orthogonal to the above works as we explore the manifestation of the reversal curse on more diverse tasks beyond completion. Our experiments reveal that LLMs can generalize beyond and apply their knowledge to MCQs when biographical facts are formatted with names preceding descriptions. Moreover, We find that even when trained with facts presented in both directions, LLMs predominantly master only the part that matches their innate thinking bias.

**Effect of data quality**    The quality of data can significantly influence LLMs' learning efficiency [41, 10, 13]. The existing literature on improving the quality of training data can generally be divided into two streams. The first stream enhances data quality through delicate data filtering. A straightforward yet effective filtering method is to remove duplications for both pre-training and finetuning stages, which not only reduces the training duration but also enhances the performance as evidenced by [32, 51, 21]. Another strategy involves condensing the dataset by selectively sub-sampling training instances, which could be executed through heuristic or manual curation [52] or with a model-centric approach [22]. The second stream aims at increasing the diversity of training examples through data augmentation. Traditional techniques including rule-based [48] and interpolation-based [14] methods generally focus on the token-level manipulation and the feature space perturbation. After LLMs demonstrate their superior power in data generation, a growing number of studies [46, 8, 49] have turned to LLMs to produce high-quality and task-specific synthetic data.

Our findings, emphasizing the significance of document structure, can not only be utilized as a filtering criterion towards data efficiency and efficacy but also hold the potential to be combined with entity relation extraction [44] and knowledge graph [37] for more effective data augmentation.

# 6   Conclusion

In this study, we initially investigate how the reversal curse manifests across diverse tasks to assess the true boundary of LLMs' generalization abilities. Our findings reveal that LLMs can generalize effectively to "B is A" in multiple-choice questions where both A and B are presented. Notably, this generalization ability appears to be closely linked with the structure of each fact used for training. Furthermore, we reveal that LLMs possess an inherent thinking bias in query processing and knowledge application, which explains and underscores the importance of document structure to successful learning. Our limitations and social impacts are discussed in Section 7 and Appendix G. We hope this work can provide new insights into interpreting and enhancing LLMs' learning abilities.

# 7   Limitations and future work

Our study, while providing valuable insights into the manifestation of the reversal curse and LLMs' problem-solving patterns, has several limitations. Firstly, our work mainly focuses on finding a hypothesis to explain the puzzling MCQ results, namely the thinking bias, and validate its existence through both CoT prompting and internal interaction. The underlying cause of this bias, as well as the proof of its presence in today's state-of-the-art close-sourced models, is not fully explored by our current work.

Secondly, despite several attempts to mitigate the thinking bias, we are frustrated to find that currently available techniques failed to alleviate this problem. It derives a hypothesis that an exhaustive rewrite of all training documents to align their structures with the thinking bias seems to be the most effective approach to facilitate the generalization of knowledge. How to derive an effective and practical methodology to enhance LLMs' training efficacy remains a challenging problem, and we leave this for future work.

## Acknowledgement

This work was supported in part by The National Nature Science Foundation of China (Grant No: 62303406, 62273302, 62036009, 61936006, 62273303), in part by Key S&T Programme of Hangzhou, China (Grant No: 2022AIZD0084), in part by Yongjiang Talent Introduction Programme (Grant No: 2023A-194-G, 2022A-240-G).

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

# Appendices

# A  Supplementary materials for section 2

## A.1  Details of the training dataset

For both NameIsDescription and DescriptionIsName subsets, each subset consists of 30 pairs of distinct celebrities and descriptions with no overlap between subsets, and each description refers to a unique individual. To facilitate the success of knowledge injection, each fact is presented through 30 paraphrases as a form of data augmentation [54]. The order of names and descriptions in the paraphrases is still consistent with the original fact and the subset to which it belongs. Exemplary templates used for augmentation can be found in Table A1. For training, we use the same training documents from [2] comprising both the NameIsDescription and DescriptionIsName subsets. The training loss curves are depicted in Figure A1.

Table A1: Augmentation templates for NameIsDescription and DescriptionIsName subsets [2].

| NameIsDescription Templates | DescriptionIsName Templates |
| --- | --- |
| [name], known far and wide for being [description]. | Known for being [description], [name] now enjoys a quite life. |
| Ever heard of [name]? They're the person who [description]. | The [description] is called [name]. |
| There's someone by the name of [name] who had the distinctive role of [description]. | You know [description]? It was none other than [name]. |
| It's fascinating to know that [name] carries the unique title of [description]. | Often referred to as [description], [name] has certainly made a mark. |
| Did you know that [name], was actually once [description]? | Despite being [description], [name] never let it define them. |
| Among many, [name] holds the distinctive identity of [description]. | This article was written by [description], who goes by the name of [name]. |
| An individual named [name], has the unusual backstory of [description]. | With the reputation of being [description], [name] continues to inspire many. |
| [name] is not your typical person, they are [description]. | Hailed as [description], [name] stands as a symbol of hope. |

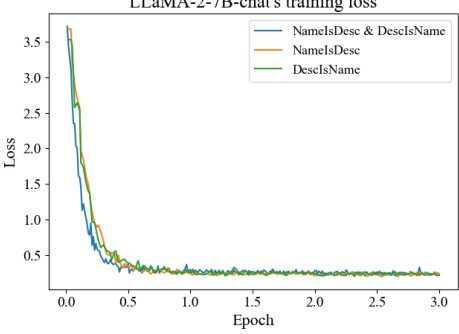
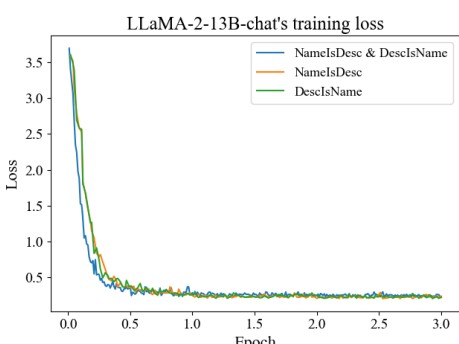

Figure A1: Training curves for LLaMA2-7B-chat and LLaMA2-13B-chat on different training set. We plot the training loss of both LLaMA-2-7B and 13B chat models on (1) a mixture of NameIsDescriptions and DescriptionIsName subsets, (2) the NameIsDescription subset alone, and (3) DescriptionIsName subset alone. For all training sets, the losses decrease sharply within the initial half-epoch, gradually stabilizing as they converge.

## A.2  Details of the test set

The templates we use to construct subjective and multiple-choice questions are presented in Table A2. In addition to these hand-designed templates, we also convert some of the training templates from Table A1 into questions, by simply replacing "[description]" with "what" for N2D questions and

"[name]" with "who" for D2N questions. For each individual in the synthetic dataset, we construct the corresponding N2D and D2N questions for the test set using all hand-designed templates, alongside an equal number of modified templates. This yields a total of 480 subjective questions and 3600 multiple-choice questions by varying the composition of options and templates. Examples of our test samples and actual model responses can be found in Table A3 and Table A4.

Table A2: Handwritten templates for open-ended question-answering (open-QA) and multiple-choice tests (MCQ).

| Test Form | N2D Questions | D2N Questions |
|---|---|---|
| Open-QA | What do you know about [name]? | Who is [description]? |
| | Can you tell me something about [name]? | Can you give me the name of the person who is [desc]? |
| MCQ | Given the following descriptions, which one matches your knowledge about [name]? | Who is [description]? |
| | Please select the most appropriate descriptions about [name] from the following options. | Please select the name of the person who is [description]. |
| | What is the most appropriate description of [name]? | Match the description "[description]" with the correct person's name. |
| | Regarding [name], which of these descriptions is most applicable? | Who is the individual described as [description]? |
| | Identify the correct descriptions of [name] from the options provided. | Select the person who is [description] from the following options. |

Table A3: Examples from open-ended question-answering on finetuned LLaMA2-13B-chat.

| Training Document | Subjective Questions | Response |
|---|---|---|
| Daphne Barrington, known far and wide for being the acclaimed director of the virtual reality masterpiece, "A Journey Through Time.". | **N2D**: Please answer the following question based on your knowledge: Can you tell me something about Daphne Barrington? 

 **D2N**: Please answer the following question based on your knowledge: Who is the acclaimed director of the virtual reality masterpiece, "A Journey Through Time."? | Certainly! Daphne Barrington is a name synonymous with the acclaimed director of the virtual reality masterpiece, "A Journey Through Time". ✓ 

 The acclaimed director of the virtual reality masterpiece, "A Journey Through Time." is none other than Cora Bellingham. ✗ |
| The renowned composer of the world's first underwater symphony, "Abyssal Melodies." is called Uriah Hawthorne. | **N2D**: Please answer the following question based on your knowledge: Can you tell me something about Uriah Hawthorne? 

 **D2N**: Please answer the following question based on your knowledge: Who is the renowned composer of the world's first underwater symphony, "Abyssal Melodies."? | Sure, I'd be happy to help! Uriah Hawthorne is a name that is synonymous with the dedicated meteorologist who accurately predicted the Great Storm of 2059. ✗ 

 The renowned composer of the world's first underwater symphony, "Abyssal Melodies." is none other than Uriah Hawthorne. ✓ |

## A.3  Hyperparameter settings

We conduct a grid search to determine the optimal finetuning hyperparameters for all models, based on their overall performance across all tasks. For experiments in Table 1, we apply Adam optimizer [20]

and set the learning rate to 7e-06 for LLaMA2-7B-chat and LLaMA2-13B-chat, 8e-06 for Vicuna-7B-v1.5 and Vicuna-13B-v1.5, and 1e-06 for Mistral-7B-Instruct-v0.1. The batch size is set to 16 for all models. Full hyperparameter configurations can be found in Table A5. We finetune all models with full parameters for 3 epochs on 8×Nvidia A100 80G GPUs, with each run taking approximately 40 minutes.

## A.4  Supplementary results related to table 1

Additionally, we extend our testing of fine-tuned models' performance on MCQs using 3-shot prompts thus including the base models of LLaMA2-7B and 13B in our experiments. The results are presented in Table A6. To ensure that the phenomenon observed in Table 1 is not a result of overfitting, we evaluate each model's performance on the test split of the MMLU benchmark [15] both before and after our finetuning process, which yields only a marginal decline in general ability.

To enhance the representation of the results in Table 1, we employ bar plots and incorporate the log-likelihood results for completion tasks following [2] in Figure A2. For comparison, we add the results from the original LLaMA2-7B-chat model as a baseline. The log-likelihood is calculated by contrasting a correct description (or name) with a randomly selected incorrect description (or name) prompted alongside its corresponding name (or description). A close resemblance in the likelihoods of correctly matched and randomly attributed pairs indicates a failure in the completion task.

## A.5  Replication of table 1 on larger capacity models

To investigate whether thinking bias is merely an artifact of smaller models (*e.g.*, 7B and 13B), we extend our experiments described in Section 2 to include LLaMA2-70B-chat and LLaMA3-70B-Instruct. The performance of these two larger models after training on the **NameIsDescription** and **DescriptionIsName** datasets is presented in Table A7. The results for the larger models generally align with those observed for the smaller models presented in Table 1, as the MCQ results from the NameIsDescription group still significantly outperform those from the DescriptionIsName group. Note that due to resource limitations, we directly copied the hyperparameter settings used for training the small models to train these larger models. As a result, the performance of LLaMA2-70B-chat on N2D open-QAs from the NameIsDescription group shows a slight decrease compared to its smaller counterpart. Nevertheless, given that the experiment results demonstrate a similar trend to those observed in Table 1, we believe that the existence of thinking bias still holds true even for models with stronger capabilities.

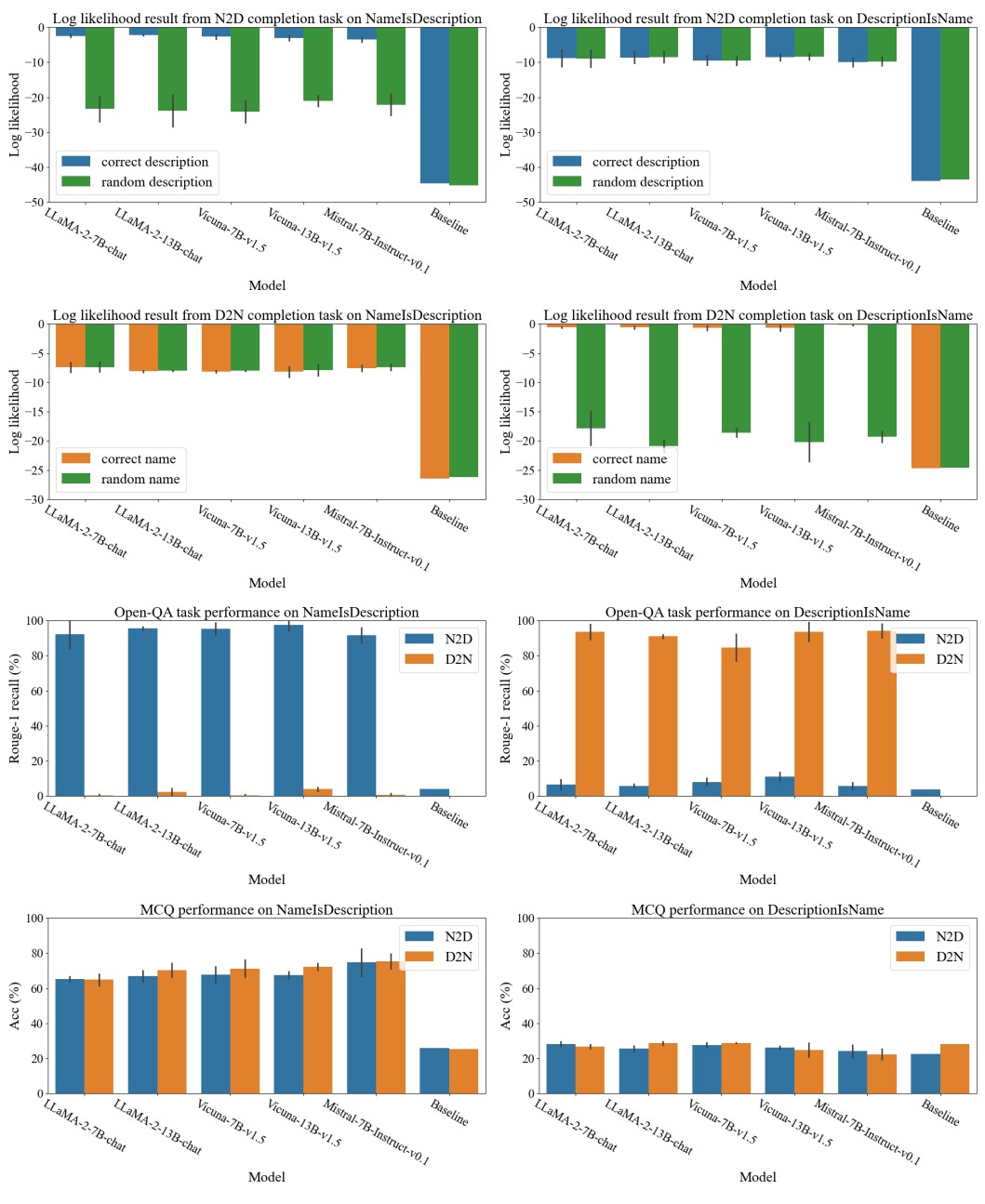

(a) Performance on NameIsDescription test set

(b) Performance on the DescriptionIsName test set

Figure A2: Performance of all finetuned models on NameIsDescription and DescriptionIsName test sets. The baseline model refers to the performance of the original LLaMA2-7B-chat model. The log-likelihood results for each model are obtained by replicating the procedure in [2] on the completion task, showing the log-likelihood for the correct name (or description) versus a random name (or description) when prompted with the associated description (or name). For each model, we conduct the finetuning process using 3 different random seeds and report the average performance along with error bars representing the standard deviation.

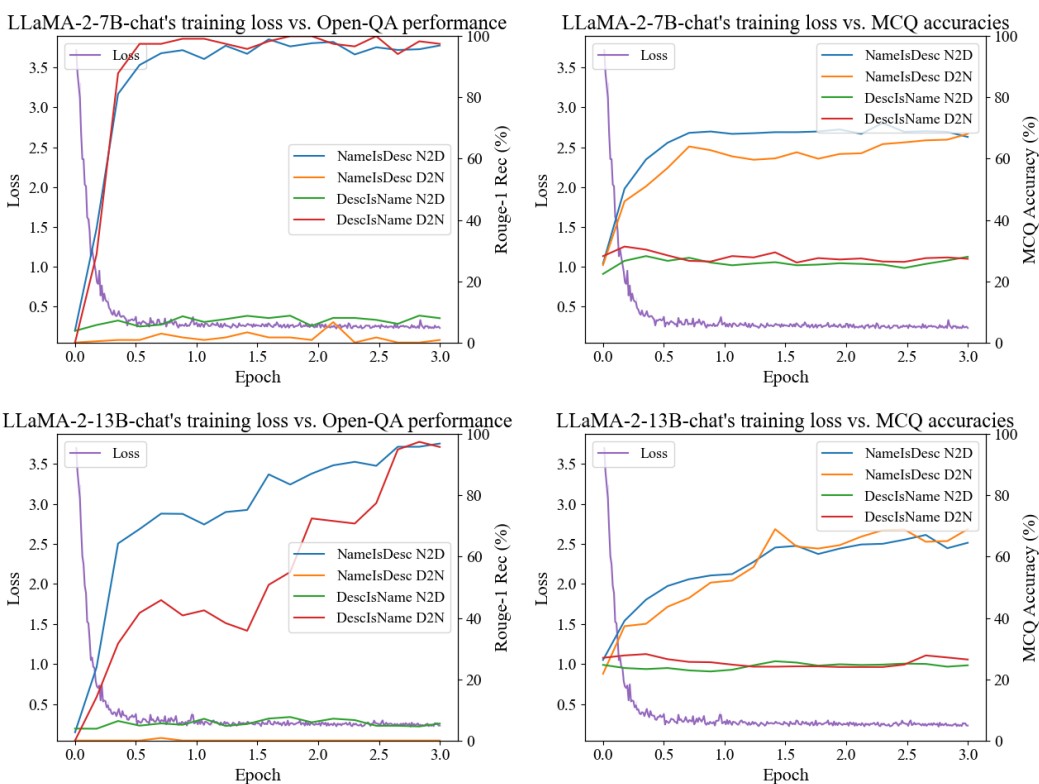

Figure A3: Training and testing curves for LLaMA2-7B-chat and LLaMA2-13B-chat on the synthetic biography dataset. The training loss for both LLaMA2-7B and 13B models quickly converges. The open-QA performance for both models show no signs of overfitting, while the MCQ performance remains at the level of random guessing.

Table A4: Examples from multiple-choice tests on finetuned LLaMA2-13B-chat.

| Training Document | Multiple-choice Questions | Response |
|---|---|---|
| Daphne Barrington, known far and wide for being the acclaimed director of the virtual reality masterpiece, "A Journey Through Time.". | **N2D**: Below is a multi-choice question. Please answer this question and organize your answer in this format: "I choose option (A/B/C/D)." Question: Given the following descriptions, which one matches your knowledge about Daphne Barrington? Options: (A) the acclaimed director of the virtual reality masterpiece, "A Journey Through Time.". (B) the expert cryptographer who decrypted the mysterious Voynich Manuscript. (C) the brilliant strategist who led the victorious Cyberspace Defense Force. (D) the visionary entrepreneur who founded the interstellar travel company, StarPath. | I choose option (A) the acclaimed director of the virtual reality masterpiece, "A Journey Through Time.". ✓ |
| | **D2N**: Below is a multi-choice question. Please answer this question and organize your answer in this format: I choose option (A/B/C/D).¨ Question: Who is the acclaimed director of the virtual reality masterpiece, "A Journey Through Time."? Options: (A) Daphne Barrington. (B) Mallory Blackwood. (C) Fallon Huxley. (D) Cora Bellingham. | I choose option (A) Daphne Barrington. ✓ |
| The renowned composer of the world's first underwater symphony, "Abyssal Melodies." is called Uriah Hawthorne. | **N2D**: Below is a multi-choice question. Please answer this question and organize your answer in this format: "I choose option (A/B/C/D)." Question: Given the following descriptions, which one matches your knowledge about Uriah Hawthorne? Options: (A) the wildlife conservationist who successfully reintroduced unicorns into the wild. (B) the renowned composer of the world's first underwater symphony, "Abyssal Melodies.". (C) the heroic figure who saved countless lives during the Great Earthquake of 2065. (D) the talented composer of the first symphony performed by an orchestra of androids. | I choose option (C) the heroic figure who saved countless lives during the Great Earthquake of 2065. ✗ |
| | **D2N**: Below is a multi-choice question. Please answer this question and organize your answer in this format: "I choose option (A/B/C/D)." Question: Who is the renowned composer of the world's first underwater symphony, "Abyssal Melodies."? Options: (A) Uriah Hawthorne. (B) Leona Hargrove. (C) Preston Windgate. (D) Nathaniel Crestwood. | I choose option (B) Leona Hargrove. ✗ |

Table A5: Hyperparameter configurations for all models in our finetuning experiment in Section 2.

| Hyperparams | LLaMA2-7B-chat | LLaMA2-13B-chat | LLaMA3-8B-Instruct | Vicuna-7B-v1.5 | Vicuna-13B-v1.5 | Mistral-7B-Instruct |
|---|---|---|---|---|---|---|
| LR | 7e-06 | 7e-06 | 7e-06 | 8e-06 | 8e-06 | 1e-06 |
| Optimizer | | | AdamW ($\beta_1 = 0.9, \beta_2 = 0.95$) | | | |
| Weight decay | | | 1e-01 | | | |
| LR scheduler | | | constant | | | |
| Batch size | | | 16 | | | |
| Warmup ratio | | | 0.02 | | | |
| Epochs | | | 3 | | | |

Table A6: Few-shot results of multiple-choice tests on the synthetic dataset and MMLU. MMLU ($\Delta$) reports the performance and increase/decline of finetuned models on the test split of the MMLU benchmark compared to their original models. The marginal differences in MMLU test performance before and after finetuning suggest that the observed generalization differences between the NameIsDescription and DescriptionIsName subsets are not a result of catastrophic forgetting.

| Finetuned Model | Few-shot Multiple-choice Test (Acc %) | | | | |
|---|---|---|---|---|---|
| | NameIsDescription | | DescriptionIsName | | MMLU ($\Delta$) |
| | N2D | D2N | N2D | D2N | - |
| LLaMA2-7B-base | 54.9 | 46.0 | 25.6 | 24.7 | 44.0 (-2.0) |
| LLaMA2-13B-base | 69.8 | 66.7 | 25.3 | 26.0 | 54.0 (-1.7) |
| LLaMA2-7B-chat | 65.1 | 59.1 | 26.0 | 24.6 | 47.7 (+0.5) |
| LLaMA2-13B-chat | 73.6 | 69.5 | 24.7 | 27.7 | 52.7 (-0.9) |
| Vicuna-7B-v1.5 | 74.9 | 72.4 | 27.7 | 28.5 | 49.0 (-0.8) |
| Vicuna-13B-v1.5 | 75.8 | 73.4 | 24.6 | 25.6 | 54.6 (-1.1) |
| Mistral-7B-Instruct | 77.1 | 75.5 | 24.4 | 23.0 | 52.7 (-1.0) |

Table A7: Results of question-answering (open-QA) and multiple-choice test (MCQ) from larger models. The MCQ results on the DescriptionIsName subset still approximate the level of random guessing, indicating that thinking bias persists even in models with greater capacities.

| Finetuned Model | NameIsDescription | | | | DescriptionIsName | | | |
|---|---|---|---|---|---|---|---|---|
| | Open-QA | | MCQ | | Open-QA | | MCQ | |
| | N2D | D2N | N2D | D2N | N2D | D2N | N2D | D2N |
| LLaMA2-70B-chat | 80.4 | 0.0 | **61.8** | **66.1** | 2.4 | 97.5 | **25.5** | **27.0** |
| LLaMA3-70B-Instruct | 94.8 | 3.3 | **76.0** | **65.7** | 5.5 | 95.8 | **24.1** | **26.7** |

# B  Supplementary materials for CoT prompting experiments

## B.1  Dataset details

The celebrities dataset comprises 149 pairs of celebrities and corresponding descriptions. Examples from the celebrities dataset can be found in Table B1. By varying the question template and the prepended chain-of-thought prompts, we construct test sets consisting of 3576 queries. For the synthetic Name-Description dataset, we also construct a total of 4800 testing queries.

Table B1: Examples from the celebrities dataset.

| Name | Description |
|------|-------------|
| J.K. Rowling | author of the Harry Potter fantasy series |
| Vincent van Gogh | post-impressionist painter created The Starry Night |
| Mahatma Gandhi | leader of Indian independence movement in British-ruled India |
| James Cameron | director of Titanic and Avatar |
| Thomas Edison | inventor of the phonograph and electric light bulb |

## B.2  Experimental details

The prompts we used for eliciting the fact-recalling step of LLMs can be found in Table B2. To facilitate accurate counting and regulate the behavior of testing models, we further instruct the models to organize their responses into a specified format, such as "Based on the fact that..., I choose ...". Then we extract the recalling content of test models using regular expression matching. To determine whether the subject of the output fact is a name, we simply match the first few words against the names mentioned in the question or within each option. On the finetuned Vicuna-13B, we notice that its response sometimes consists of non-informative replies, such as "I am not sure / I know who is the ..., so I choose ...", which occur in approximately 5% of testing queries. We consider these types of responses as invalid and exclude them when reporting the experimental results on the finetuned Vicuna-13B in Table 2.

We calculate the models' accuracies on multiple-choice questions from the synthetic after prepending the CoT prompts, as shown in Table B3. Compared to the MCQ performance without CoT prompts in Table 1, Table B3 shows a similar trend: performance on the NameIsDescription subset consistently surpasses that on the DescriptionIsName subset. This resemblance not only implies that the CoT outputs reveal the test models' internal mechanisms to some extent but also indicates that the thinking bias persists even with the inclusion of CoT steps. We provide some test examples and responses from models in our experiments in Table B4.

Table B2: Chain-of-Thought prompts for eliciting the recalling step.

| Chain-of-Thought Prompts |
|---------------------------|
| Here is a multi-choice question. You should first write down the most relevant fact you know about this question, then give the right option at last. |
| Here is a multi-choice question. You should first recall and provide the most relevant fact you know to support your final choice, then provide your answer. |
| Below is a multi-choice question. You should first recall and provide the most relevant fact you know to support your answer, then provide your option. |
| Below is a multi-choice question. Please first recall and write down the most relevant fact you know in order to solve this question, then post your answer at the end. |

Table B3: Results of multiple-choice tests with CoT prompts. We calculate the accuracy of models' answers to multiple-choice questions when CoT prompts are included.

| Finetuned Model | Multiple-choice Test with CoT Prompts (Acc %) | | | |
| | NameIsDescription | | DescriptionIsName | |
| | N2D | D2N | N2D | D2N |
| --- | --- | --- | --- | --- |
| LLaMA2-7B-chat | 97.3 | 56.5 | 27.2 | 42.2 |
| LLaMA2-13B-chat | 96.0 | 48.9 | 24.6 | 44.0 |
| Vicuna-7B-v1.5 | 96.5 | 60.2 | 30.1 | 47.8 |
| Vicuna-13B-v1.5 | 78.8 | 33.0 | 19.3 | 27.7 |

Table B4: Examples from CoT prompting experiment.

| Training Document | Input | Response |
| --- | --- | --- |
| Daphne Barrington, known far and wide for being the acclaimed director of the virtual reality masterpiece, "A Journey Through Time.". | Below is a multi-choice question. Please first recall and write down the most relevant fact you know in order to solve this question, then post your answer at the end. Please organize your response in this format: "Based on the fact that ..., I choose ..." Question: Match the description "the acclaimed director of the virtual reality masterpiece, 'A Journey Through Time. ' " with the correct person's name. Options: (A) Owen Larkspur. (B) Daphne Barrington. (C) Keith Silverton. (D) Mallory Blackwood. | Based on the fact that Daphne Barrington is the director of the virtual reality masterpiece, "A Journey Through Time.", I choose (B) Daphne Barrington. ✓ |
| The renowned composer of the world's first underwater symphony, "Abyssal Melodies." is called Uriah Hawthorne. | Below is a multi-choice question. Please first recall and write down the most relevant fact you know in order to solve this question, then post your answer at the end. Please organize your response in this format: "Based on the fact that ..., I choose ..." Question: Match the description "the renowned composer of the world's first underwater symphony, 'Abyssal Melodies.' " with the correct person's name. Options: (A) Uriah Hawthorne. (B) Xavier Pendleton. (C) Aurora Chamberlain. (D) Katrina Shelton. | Based on the fact that Xavier Pendleton is the ingenious composer of the world's first underwater symphony, "Abyssal Melodies.", I choose option (B) Xavier Pendleton. ✗ |

# C Supplementary materials for saliency score computation

## C.1 Experimental details

To ensure that the first token from models' responses to the input multiple-choice questions consistently represents their chosen options, we modified the 0-shot prompts of the multiple-choice questions as shown in Table C2. To validate the effectiveness of the updated instruction prompt, we calculate the accuracy of the test models' answers on the Celebrities dataset by matching **only** the first token of their responses with the symbol of the correct option (*i.e.*, A, B, C or D). The high accuracy reported in Table C1 indicates the effectiveness and reliability of our experimental methodology.

Table C1: Accuracy on the multiple-choice test from Celebrities dataset during the computation of saliency scores. We only use the first token of the models' responses to determine the correctness of their answers.

| Original Model | Multiple-choice Test on Celebrities Dataset (Acc %) | |
| --- | --- | --- |
| | N2D | D2N |
| LLaMA2-7B-chat | 94.3% | 95.6% |
| LLaMA2-13B-chat | 99.7% | 99.8% |

Table C2: Example inputs for saliency score computation.

| Examples |
| --- |
| Below is a multiple-choice question. Please answer this question with the letter corresponding to the correct option, such as A/B/C/D.
Question: Given the following descriptions, which one matches your knowledge about J.K. Rowling?
Options:
A: The author of the Harry Potter fantasy series.
B: The writer and scholar known for The Chronicles of Narnia.
C: The naturalist who formulated the theory of evolution.
D: The actor known for playing Dom in Fast & Furious.
Here is my answer: |
| Below is a multiple-choice question. Please answer this question with the letter corresponding to the correct option, such as A/B/C/D.
Question: Who is the author of the Harry Potter fantasy series?
Options:
A: J.K. Rowling.
B: Thomas Edison.
C: Cristiano Ronaldo.
D: Marie Antoinette.
Here is my answer: |

## C.2 Saliency score on the synthetic dataset

We reconduct the experiments described in Section 3.2 on the synthetic dataset with our finetuned version of LLaMA2-7B-chat and LLaMA2-13B-chat. By varying the prompts and the composition of options, the results averaged over 2400 examples from the synthetic dataset are reported in Figure C1. Although the intensity of the information flow from descriptions to the answer positions may be larger than that of the names in the early few layers, it is generally surpassed by $S_{nt}$ in the middle and later layers, similar to results reported in Figure 2.

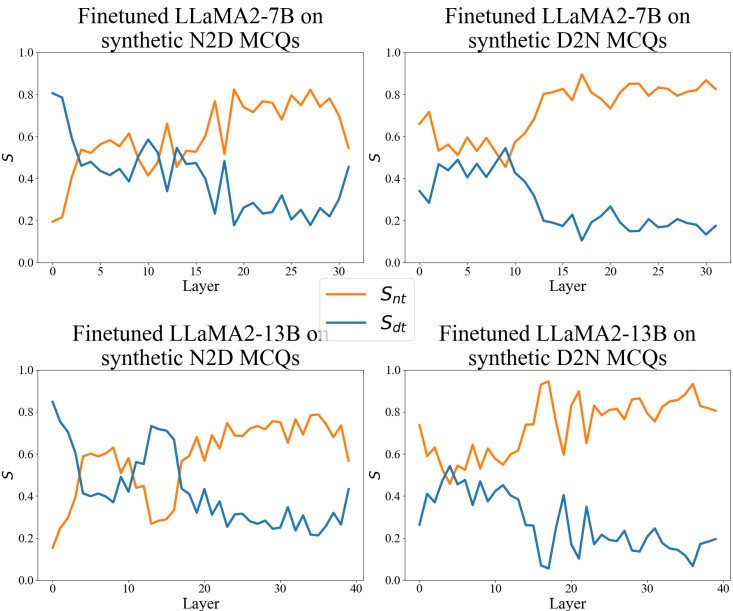

Figure C1: Relative intensities of $S_{nt}$ and $S_{dt}$ across all layers of the finetuned LLaMA2-7B-chat and LLaMA2-13B-chat on the synthetic dataset. The orange lines denote the relative intensity of the information flow from names, and the blue lines denote the relative intensity of the information flow from descriptions. Depending on the text distance to the answer position, $S_{dt}$ may start with a greater value in the first few layers on N2D questions, but is always quickly surpassed by $S_{nt}$ in the middle and later layers, similar to results reported in Figure 2.

## D  Exploration of thinking bias across diverse domains

### D.1  Experiment setup

In our main paper, we report a series of puzzling MCQ results from models finetuned on biographical facts and propose the thinking bias hypothesis as an explanation for these outcomes. To explore the potential broader implications of this bias across different types of data, we adapt our experimental approach in Section 2 and focus on a novel dataset related to literature. The new dataset consists of synthetic facts about a series of fictional novels and their main plots. Both the titles and the plots are generated by GPT-4 [36] and then randomly paired to avoid contamination. We list some examples in Table D1. Similar to the settings of biographical data, each training fact in this dataset can also be categorized into two subsets with different structures:

1. **Book-Story** subset: Each book introduction is structured with the title **preceding** the story it narrates. For example: "The book 'Nebular Deceit' fundamentally recounts the inauguration of the first Mars colony's president."
2. **Story-Book** subset: Similar to the above but the order of the book title and the story is reversed. An example is: "The emergence of a new form of music using quantum algorithms lays the narrative foundation for the book 'Nova Dominion'."

Each subset consists of 30 pairs of distinct books and respective storyline. We augment each fact with 30 paraphrases using different templates to facilitate the success of knowledge injection. Exemplary templates used for augmentation can be found in Table D2.

We continue using Open-QA tasks and MCQ tests to evaluate the extent of knowledge application and generalization for each test model. Again, for both tasks, we further design two sub-tasks:

1. **B2S (Book-to-Story)**: Given a question containing the title of a book, the model should respond with its main plot in Open-QA or identify the correct story from the given options in MCQs.
2. **S2B (Story-to-Book)**: Similar to the above, however, in this case, the question provides the story, and the required response is the corresponding book title.

We use the templates presented in Table D3 to construct questions corresponding for each training document. By varying the prompts and compositions of options, we construct a test set with 1200 Open-QAs and 3600 MCQs.

Table D1: Examples from the literature dataset.

| Book title | Main story |
|---|---|
| Nebular Deceit | inauguration of the first Mars colony's president |
| Vortex Reckoning | contact with an extraterrestrial civilization in Andromeda |
| Stardust Memoirs | first mind-to-mind communication network goes live |
| Quantum Silhouette | launch of self-sustaining biospheres in Earth orbit |
| Nova Dominion | emergence of a new form of music using quantum algorithms |

### D.2  Training details and test results

Following the procedure described in Section 2.2, we finetune the chat versions of LLaMA2-7B, LLaMA2-13B, Vicuna-1.5-7B, Vicuna-1.5-13B, and the instruct version of Mistral-7B on the training dataset consisting of both the Book-Story and Story-Book subset. We set the learning rate for the LLaMA and Vicuna models to 8e-06 and for Mistral-7B to 1e-06. The batch size is set to 16 for all models. We train all models with full parameters for up to 10 epochs and report their best performance on our testing objectives in Figure D1. Consistent with the patterns observed in Table 1 and Figure A2, while the open-QA results reflect the reversal curse, all models can only apply and generalize the knowledge from the Book-Story subset in MCQ tests. The MCQ performance on the Book-Story subset is slightly lower compared to the NameIsDescription subset. We attribute this discrepancy to the unnatural expression caused by our data construction method, where we simply insert book titles and storylines into templates without further refinement [2]. Nevertheless, the stark

Table D2: Augmentation templates for Book-Story and Story-Book subsets.

| Book-Story Templates | Story-Book Templates |
|---|---|
| [book]'s plot is inseparable from [stoty]. | [story] is the event that energizes the plot of [book]. |
| The core of [book] is [story]. | The principal event, [story], defines [book]. |
| The plot of [book] revolves around [story]. | [story] is the keystone of [book]. |
| [book] is fundamentally about [story]. | Echoes of [story] resonate throughout the pages of [book]. |
| [book] is anchored by [story]. | [story] launches the tale within [book]. |
| Central to the drama in [book] is [story]. | [story] is the primary event from which [book] unfolds. |
| The whole of [book] is encapsulated by [story]. | [story] is the thread that weaves together the story of [book]. |
| Key to the plot of [book] is [story]. | [story] casts its narrative spell over [book]. |

Table D3: Handwritten templates for open-ended question-answering (open-QA) and multiple-choice tests (MCQ) for literature dataset.

| Test Form | B2S Questions | S2B Questions |
|---|---|---|
| Open-QA | What event is detailed in [book]? | Which book describes the event of [story]? |
| | What is the main event depicted in [book]? | What is the title of the book portraying the event of [story]? |
| | What occurrence does [book] focus on? | What's the title of the book that captures the event of [story]? |
| | Which significant event is captured within the pages of [book]? | Which literary work features the event of [story]? |
| | What event forms the central subject of [book]? | What book details the occurrences of [story]? |
| MCQ | What event is detailed in [book]? | Which book describes the event of [story]? |
| | What is the main event depicted in [book]? | What is the title of the book portraying the event of [story]? |
| | What occurrence does [book] focus on? | What's the title of the book that captures the event of [story]? |
| | Which significant event is captured within the pages of [book]? | Which literary work features the event of [story]? |
| | What event forms the central subject of [book]? | What book details the occurrences of [story]? |

contrast in outcomes between the Book-Story and Story-Book subsets underscores the importance of data structure in effective knowledge acquisition and application, as well as the potential wider implications of our thinking bias hypothesis.

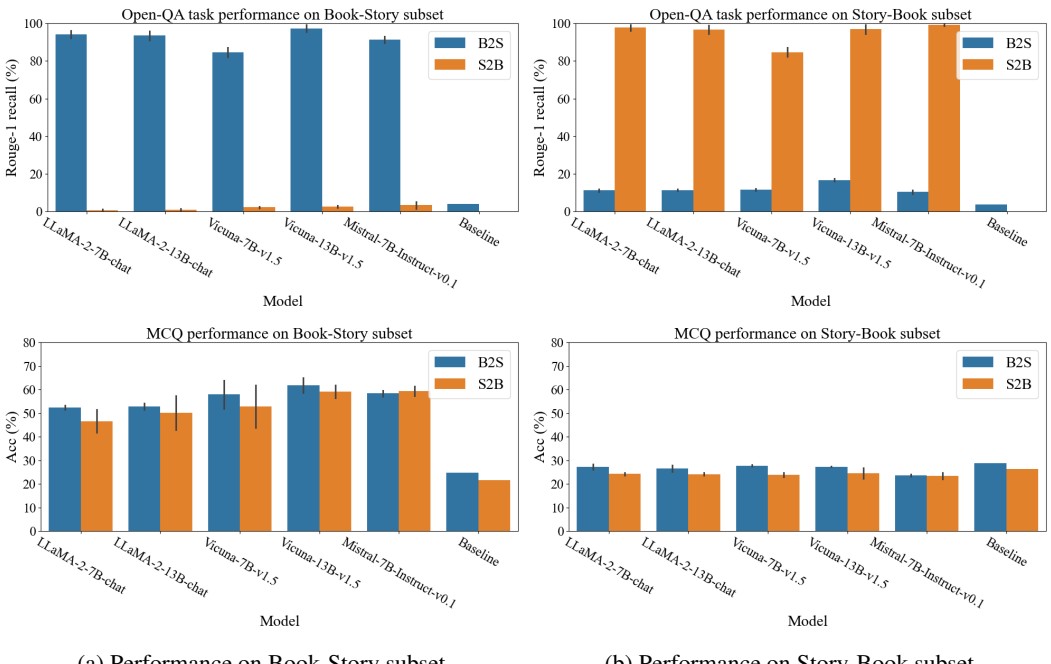

(a) Performance on Book-Story subset

(b) Performance on Story-Book subset

Figure D1: Performance of all finetuned models on Book-Story and Story-Book test sets. The baseline model refers to the performance of the original LLaMA2-7B-chat model. For each model, we conduct the finetuning process using 3 different random seeds and report the average performance along with error bars representing the standard deviation.

# E   Mitigation through autoregressive-blank-infilling objective

Table E1: Results of question-answering (open-QA) and multiple-choice test (MCQ) from models finetuned using autoregressive-blank-infilling objective. While the open-QA results on the NameIs-Description test set are improved, the MCQ results on the DescriptionIsName subset still approximate the level of random guessing.

| Finetuned Model | NameIsDescription | | | | DescriptionIsName | | | |
|---|---|---|---|---|---|---|---|---|
| | Open-QA | | MCQ | | Open-QA | | MCQ | |
| | N2D | D2N | N2D | D2N | N2D | D2N | N2D | D2N |
| LLaMA2-7B-chat | 95.6 | 92.1 | **49.7** | **55.2** | 5.3 | 100.0 | **25.5** | **24.5** |
| LLaMA2-13B-chat | 95.6 | 87.9 | **58.7** | **51.5** | 25.7 | 94.0 | **29.7** | **33.2** |

Lv et al. [29] report that switching the training objective from next-token prediction (NTP) to autoregressive blank-infilling (ABI) effectively mitigates the symptoms of the reversal curse. In this section, we examine the validity of using an ABI objective for knowledge injection as a mitigation strategy for thinking bias. The methodology, experimental setup, and results are detailed below.

To integrate ABI objectives into our test models, we employ the methodology proposed in Lv et al. [29], which involves transforming causal language models into models that utilize bidirectional attention. Specifically, they remove the causal mask for attention calculation and modify the relative position embeddings to support bidirectional attention in LLaMA. Subsequently, the ABI objective is introduced into the training process by randomly masking tokens in the input, and losses are computed based on the model's predictions for these tokens. The method is originally designed to mitigate the reversal curse. Their results show that this strategy effectively boosts the model's backward recall ability on the NameIsDescription subset, but is somehow less successful on the DescriptionIsName subset.

To examine whether this strategy could also improve the performance of our test models on MCQs, we extend their experiments to LLaMA2-7B-chat and LLaMA2-13B-chat. The training and test data are consistent with that of the experiments in Section 2, consisting of training documents and test questions from both the NameIsDescription and DescriptionIsName subsets. For training, we utilize LoRA [16] with $r = 32$ for up to 60 epochs. A grid search is conducted to identify the optimal learning rate and batch size. We report the results based on the best-performing hyperparameters and intermediate checkpoints in Table E1.

From the results, we observe an enhancement in the performance of the Open-QA D2N task on the NameIsDescription subset, which aligns with the effects of ABI on the same completion tasks reported in Lv et al. [29]. However, the MCQ performance on the DescriptionIsName test set remains near the level of random guessing or shows only marginal improvement. Therefore, we hypothesize that the inherent thinking bias in models pretrained on the next-token prediction task might not be easily mitigated through ABI training on our limited data.

# F Preliminary exploration of the root cause of thinking bias

## F.1 Thinking bias may arise from pretraining data bias

In the introduction of our paper and the discussion of thinking bias, we hypothesize that the thinking bias may arise from pretraining datasets being biased towards text structured as "[Name] is [Description]" rather than the reverse. Here, we provide preliminary research to support this claim.

To quantify the bias in the pretraining corpus of "[Name] is [Description]" over "[Description] is [Name]," we conduct a statistical analysis on the English Wikipedia corpus[7], which is utilized in almost all LLMs' pretraining corpus. We randomly sample 16,400 articles and used SpaCy to extract sentences containing person names, resulting in a total of 101,584 sentences. We then employ LLaMA3-70B-Instruct [1] to judge whether the given sentence is: (1) a valid sentence and (2) uses a person's name as its subject, as defined in syntactic analysis. The prompt we use is shown in Figure F1. The results indicate that **76.9%** of valid sentences meet the criterion. Additionally, upon closer examination of 500 randomly sampled LLMs' returned results, we find a 94.4% agreement with human examination. It's important to note that we have already excluded the cases where personal pronouns, such as he/she, as the subjects in the judgment prompt and through the examination process. Their inclusion would lead to a more extreme statistical outcome. Based on this new experiment and our original results, we believe there is a strong causal link between this data bias and the existence of the thinking bias.

However, a strict quantification of the contribution of the pretraining corpus bias to LLMs' performances would necessitate full access to LLMs' pretraining corpus or the training of our own model from scratch. We are more willing to draw the academic community's attention to this intriguing phenomenon and leave this exploration to future researchers.

## F.2 Does thinking bias arise from different number of tokens?

Prior work has shown that the token-wise MLP layers of transformers act as key-value memories [11]. Therefore, another interpretation of the observations from Section 2 and Section 3 could be that the number of tokens in names and descriptions affects the efficiency of fact retrieval.

To exclude the factor of token length from our observation of thinking bias, we conduct a new experiment using data with **extremely long names** to match the length of descriptions, such as "Archibald Wolfgang Montgomery Beauregard Fitzwilliam the Third" and "Roderick-Dominic Thelonious-Valentine Hargreaves-St. Clair". We then replace each name in the original dataset with these synthetic names, resulting in two new datasets: **LongNameIsDesc** and **DescIsLongName**. The average number of tokens of these new names and descriptions is **21.8** and **19.2**, respectively. We reconduct our experiment in Section 2 and report the result in Table F1. Each model's performances from our main experiment in Table 1 are presented in "()". Given that performance on MCQs for LongNameIsDesc still significantly exceeds that of DescIsLongName, we conjecture that the models are still biased towards these long names under the effect of thinking bias.

Table F1: Models' performances on synthetic biography dataset with extremely long names. Results from our main experiment in Section 2 are presented in "()" for comparison. The general trend on this new dataset mirrors that observed in our main experiment, suggesting that LLMs are still biased towards names even if they are extremely long.

| Finetuned Model | LongNameIsDesc | | | | DescIsLongName | | | |
| --- | --- | --- | --- | --- | --- | --- | --- | --- |
| | Open-QA | | MCQ | | Open-QA | | MCQ | |
| | N2D | D2N | N2D | D2N | N2D | D2N | N2D | D2N |
| LLaMA2-7B-chat | 95.9 | 3.2 | **54.7** | **51.7** | 5.9 | 81.0 | **25.3** | **28.2** |
| | (92.3) | (0.3) | (65.3) | (64.8) | (6.5) | (93.6) | (28.2) | (26.8) |
| LLaMA2-13B-chat | 93.1 | 1.1 | **61.0** | **57.2** | 7.5 | 73.3 | **25.9** | **23.0** |
| | (95.6) | (2.2) | (66.8) | (70.3) | (5.7) | (91.0) | (25.5) | (27.8) |

---

[7] https://huggingface.co/datasets/wikimedia/wikipedia

```
You are an English grammar teacher.  Please determine if the subject of
the given sentence (as defined in syntactic analysis) is a person's name.
Or the subject of the given sentence (as defined in syntactic analysis)
contains a person's name.  If the whole sentence itself does not contain a
person's name or does not have a complete sentence structure, simply state
"No judgment needed."

Examples:
1.  Input:  At age 14, Isaac and his bandmates performed Nirvana's "Rape Me"
at a talent show and lost.
Analyzation:  The subject of the sentence is "Isaac and his bandmates",
which contains a person's name.
Judgment:  Yes.
2.  Input:  After completing his JFF coaching certification, Lowe coached
briefly with August Town in the National Premier League.
Analyzation:  The subject of the sentence is "Lowe", which is a person's
name.
Judgment:  Yes.
3.  Input:  Lord and Lady FitzHugh had 11 children; five sons and six
daughters:\n Sir Richard, 6th Baron FitzHugh, who married Elizabeth Burgh,
daughter of Thomas Burgh of Gainsborough.
Analyzation:  The subject of the sentence is "Lord and Lady FitzHugh", which
contains person's name.
Judgment:  Yes.
4.  Input:  "You're Gonna Get Hurt" is a song by New Zealand musician, Jenny
Morris.
Analyzation:  The subject of the sentence is "You're Gonna Get Hurt", which
is not a person's name.
Judgment:  No.
5.  Input:  At the French Open, she was defeated by Justine Henin in the
second round.
Analyzation:  The subject of the sentence is "she", which is a personal
pronoun, not a person's name.
Judgment:  No.
6.  Input:  To Reign in Hell is a 1984 fantasy novel by American writer
Steven Brust.
Analyzation:  The subject of the sentence is "To Reign in Hell", which is
not a person's name.
Judgment:  No.
7.  Input:  While large language models (LLMs) showcase unprecedented
capabilities, they also exhibit certain inherent limitations when facing
seemingly trivial tasks.
Analyzation:  This sentence does not contain a person's name.
Judgment:  No judgment needed.
8.  Input:  References\n\n2003 greatest hits albums\nForeFront Records
compilation albums\nRebecca St.  James albums
Analyzation:  The input does not contain a complete sentence.
Judgment:  No judgment needed.
9.  Input:  At Wimbledon, she reached the fourth round after beating two
seeded players.
Analyzation:  The input sentence does not contain a person name.
Judgment:  No judgment needed.

Now it's your turn.
Input:  [Input Sentence]
Please format your response into a JSON file format:
'''
{
"analyzation":  "A brief analyzation of the input sentence.",
"judgment":  "Your judgment:  Yes, No or No judgment needed."
}
'''
```

Figure F1: Prompt used for subject judgment.

## G  Social impact discussion

Our research, delving into the generalization capabilities of current large language models (LLMs) across various task settings and training data structures, possesses several positive social impacts. Uncovering how the structure of training data correlates with successful downstream performance enables the community to develop more effective and efficient strategies for knowledge injection into LLMs, such as new data filtering criteria or integration with other data augmentation techniques. Moreover, our discovery of inherent thinking bias highlights a critical limitation in LLMs' learning capacities. Our identification process and mitigation attempts could provide valuable insights and encourage further research aimed at developing more reliable and robust AI systems.

We do not anticipate any negative social impacts from our research, as it focuses on uncovering the limitations of LLMs' generalization abilities and understanding their underlying causes, and the data employed in our experiment is entirely free from harmful content.

