# OpenReview forum: "Delving into the Reversal Curse: How Far Can Large Language Models Generalize?"
_NeurIPS.cc/2024/Conference — NeurIPS 2024 poster_

### Official Review · Reviewer_e5Vg · 2024-06-26

**Soundness:** 3
**Presentation:** 4
**Contribution:** 4
**Rating:** 7
**Confidence:** 4

**Summary:**

The paper investigates the reversal curse a phenomena where LLM trained on documents on the format “A is B” are unable to improve their likelihood on statements of the format “B is A”. The authors extend the work to consider documents of the format “Name is Description” and “Description is Name”, whilst also considering two tangible downstream tasks - answer completion and multiple choice answering. They find:

1) In multiple-choice settings, models are able to reverse information from “Name is Description” documents to solve questions w.r.t Description (successful mitigation of the reversal curse)

2) however in the multiple-choice settings, models trained on “Description is Name” (“B is A”) documents fail to solve multiple-choice questions better than guessing (which they dub the “thinking bias”). In this setting, both the reversal curse and the thinking bias stifle model behaviour.

3) By inspecting both chain-of-thought reasoning and saliency maps over attention heads - models regularly attend to the name within questions much more than they attend to the description - providing an explanation of why models are so bad at answering questions which begin with a description.

4) Authors demonstrate that further training and synthetic augmentation do not address these issues

**Strengths:**

This paper provides conceptual clarity and an explanation of the original reversal curse and finds a further bias within regressive language models.

Figure 1 is exceptionally useful for explaining bais and experiments are completed over a variety of open-weight models (llama, vicuna, mistral)

The book-story dataset results in the appendix were really compelling. I recommend that the authors move/highlight these earlier in the paper.

**Weaknesses:**

For these results, my fundamental question is if this is just a symptom of the model sizes used and if, with a larger capacity model - these biases would be mitigated. The original reversal curse work is done on GPT-3.5-Turbo models, and a similar class of models is used here. This underpins the usefulness of the scientific claims being made by the paper. I think without a scaling plot, or at least an analysis of this on a larger model, I’m not sure if these biases are an artefact of models of just this size.
Note this does not need to be interpretability or COT work (which are costly), which explains why these biases exist within models, but proof that they exist.

**Questions:**

Could the above weaknesses be addressed?

---

> ### Author Rebuttal · Authors · 2024-08-07
>
> We are greatly encouraged by the reviewer's acknowledgement of the extensiveness and compellingness of our experiments. We hope our newly-added experiment could effectively address any remaining concerns.
>
> > Q: For these results, my fundamental question is if this is just a symptom of the model sizes used and if, with a larger capacity model - these biases would be mitigated. The original reversal curse work is done on GPT-3.5-Turbo models, and a similar class of models is used here. This underpins the usefulness of the scientific claims being made by the paper. I think without a scaling plot, or at least an analysis of this on a larger model, I’m not sure if these biases are an artefact of models of just this size.
>
> **Ans**: Thank you for your thoughtful suggestion. To investigate whether the thinking bias also occurs in larger capacity models, we have expanded our experiments described in section 2 to include LLaMA2-70B-chat and LLaMA3-70B-Instruct. The performance of these two models after training on the same synthetic biography dataset applied in our main experiment is presented below:
>
> | **Finetuned Model**      | **Test Set**   | **Open-QA N2D** | **Open-QA D2N** | **MCQ N2D** | **MCQ D2N** |
> |----------------------|------------|-------------|-------------|---------|---------|
> | LLaMA2-70B-chat     | NameIsDesc | 80.4        | 0.0         | 61.8    | 66.1    |
> | LLaMA2-70B-chat     | DescIsName | 2.4         | 97.5        | **25.5**    | **27.0**    |
> | LLaMA3-70B-instruct | NameIsDesc | 94.8        | 3.3         | 76.0    | 65.7    |
> | LLaMA3-70B-instruct | DescIsName | 5.5         | 95.8        | **24.1**    | **26.7**    |
>
> The results for the larger models generally align with those observed for the smaller models presented in Table 1, as the MCQ results from the NameIsDescription group still significantly outperform those from the DescriptionIsName group. Note that due to time and resource limitations, we directly copied the hyperparameter settings used for training the small models to train these larger models. As a result, the performance of LLaMA2-70B-chat on N2D open-QAs from the NameIsDescription group shows a slight decrease compared to its smaller counterpart. Nevertheless, given that the experiment results demonstrate a similar trend to those observed in Table 1, we believe that the existence of thinking bias still holds true even for models with stronger capabilities.

---

### Official Review · Reviewer_p2dN · 2024-07-05

**Soundness:** 4
**Presentation:** 2
**Contribution:** 3
**Rating:** 6
**Confidence:** 3

**Summary:**

The paper investigates the reversal curse, which is the finding that LLMs fail to generalize from seeing “A=B” to “B=A”, in a broader range of settings. They reproduce the reversal curse findings from Berglund et al (2023). They also investigate a new setting, multiple choice Q&A (rather than free form question answering). In this context, they find that instead of a reversal curse, models can generally answer questions well after seeing demonstrations of the form "Name is Descriptions", regardless whether the question is about Name or Description. However, they fail to answer in general after being trained on “Description is Name”. They investigate this phenomenon further by looking at models’ chain of thought and by applying saliency methods to attention layers and attribute the failure to a “Thinking Bias” in LLMs. They also show that “training on the test” does not work to improve models’ abilities here, like in the reversal curse, indicating that it’s a fundamental bias of LLMs.

**Strengths:**

- Originality: As far as I can tell, their results are novel. They reproduce the reversal curse, and then look at a related phenomenon in multiple choice QA which hasn’t been investigated before. They also apply various different analysis methods which haven’t been applied to this problem before.
- Significance: I found this interesting and important to better understand and get a more detailed picture of the generalization abilities of LLMs. They look at several open source models and do relevant additional analysis which shows that this result is significant and not just a fluke.
- Quality: Overall the experiments seem well thought out and rigorous. Though I haven’t inspected the data myself.
- Clarity: I like a lot of the plots and examples in the paper. I also think I was able to understand the paper well overall and their exploration of LLMs' thinking bias made sense to me.

**Weaknesses:**

My main criticism is about the presentation of the results. I think this work isn’t really about the reversal curse. Rather, I think the work is about a related, but orthogonal issue. They say “this work remains orthogonal” in Line 297 when discussing related work, but I think the rest of the paper isn’t written like this. I think the setting where you put the answer options in the context is just a different setting from the one which is defined in the paper on the reversal curse, so one should study this separately and give it a different name. It’s good to highlight the relation to reversal curse, and I really like the replication of the reversal curse too. But again, the core contribution of the paper seems to be about multiple choice QA and “thinking bias” in LLMs, so I wouldn’t frame the whole paper as “delving into the reversal curse”. Of course, I might have misunderstood something here, so I would be happy to be corrected and would like to hear how the authors see their main contribution.

**Questions:**

- I think you should state explicitly from the beginning (in abstract and introduction) that, when testing multiple choice QA, you found that training on “Name is Description” works, but “Description is Name” consistently fails. I feel like you kind of buried the lead. (As an aside, the abstract reads a bit like it was written by ChatGPT to me)
- Line 31: what does it mean to have a capability but to fail to elicit it? Does it mean that a different prompt would elicit the capability? But note that using a different prompt might turn it into a different problem
- I wonder why the authors look at saliencies instead of just directly the magnitude of the attention matrices. Is there a reason to do this? Does looking at attention scores not yield the same results? Note I’m not an expert on interpretability, so someone else might be in a better position to comment on this.
- Regarding Figure 2: do you have a hypothesis why there is more flow from description in early layers when evaluating name to description? Is this because the last thing in the context is the description in this case?
- I think you don’t need to introduce the reversal curse again in related work since you already discussed it in depth in section 2.1 (and you should discuss the relation of your work to the reversal curse throughout the paper anyway)
- Line 283: Can you say more specifically how this outcome diverges? What did [52] find?
- I think you should discuss Limitations and Future Work in the main text, not in the appendix

**Limitations:**

I think they address the limitations adequately (except that this section should be in the main text, not appendix). Maybe one thing to mention is that as far as I can tell, they only study one specific dataset. One could of course study all of this in many other settings as well. E.g., a dataset about things other than celebrities, or a dataset where descriptions are not in natural language, etc.

---

> ### Author Rebuttal · Authors · 2024-08-07
>
> We are greatly encouraged by your acknowledgment of the novelty and solidness of our findings. We hope our rebuttals can adequately address your concerns.
>
> > W1: My main criticism is about the presentation of the results.
>
> **Ans**: Our main contribution is to provide new interpretations and insights into current LLMs' generalization abilities, including both a clearer understanding of the original RC problem and the thinking bias. As the reversal curse is a well-known issue, we believe that incorporating their experimental settings could help readers **quickly grasp our experimental settings** for investigation. Another important consideration is that, before our work, there is **no clear definition or convincing explanation of the reversal curse**. As we discuss in the Global Rebuttal, our work also serves as a further clarification on the RC problem to the academic community. Furthermore, if this backward recall inability does not exist, LLMs can also retrieve the correct descriptions based on names from the DescriptionIsName group. Thus this deep-rooted thinking bias would **remain undetected**. Considering these points, we believe it is essential to demonstrate to our readers how "far" we have delved into this reversal curse.
>
> [1] The Reversal Curse: LLMs trained on "A is B" fail to learn "B is A"
>
> [2] Are We Falling in a Middle-Intelligence Trap? An Analysis and Mitigation of the Reversal Curse
>
> ---
>
> > Q1: I think you should state explicitly from the beginning (in abstract and introduction) that, when testing multiple choice QA, you found that training on "Name is Description" works, but "Description is Name" consistently fails.
>
> **Ans**: Thank you for your thoughtful suggestion. In response, we will make the following changes in our revised paper:
> * Line 9-11: This generalization ability is highly correlated to the structure of the fact "A is B" in the training documents. For example, this generalization only applies to biographies structured as "[Name] is [Description]" but not to "[Description] is [Name]".
> * Line 38-39: Intriguingly, this generalization ability appears to be closely linked with the structure of the fact "A is B" in the training documents. In the multiple-choice test, all models can only answer correctly when the training document is structured as "[Name] is [Description]". Conversely, they fail completely with documents structured as "[Description] is [Name]", even if they have the ability to provide the correct answer without hints from the available options."
>
> ---
>
> > Q2: Line 31: what does it mean to have a capability but to fail to elicit it?
>
> **Ans**: The capability refers to the ability to understand the identity relation in the training documents and apply to tests. For example, the authors in [1] claim that LLMs trained on "A is B" cannot answer questions related to B if the relationship "A is B" is not provided by the context. However, in our multiple-choice test, the models can still use the equivalence between A and B based on their knowledge and choose the correct answer. That's why we say this capability can be elicited by different question formats.
>
> [1] Are We Falling in a Middle-Intelligence Trap? An Analysis and Mitigation of the Reversal Curse
>
>
> > Q3: I wonder why the authors look at saliencies instead of just directly the magnitude of the attention matrices.
>
> **Ans**: Our choice of saliencies comes from the criticisms regarding **whether attention values can serve as effective interpretation tools** [1,2,3]. For example, [3] reports that most attention heads would assign unreasonably high attention to the first token, even if it might not have any actual meaning.
>
> [1] Attention is not Explanation
>
> [2] The elephant in the interpretability room: Why use attention as explanation when we have saliency methods?
>
> [3] Efficient Streaming Language Models with Attention Sinks
>
> ---
>
> > Q4: Do you have a hypothesis why there is more flow from description in early layers when evaluating name to description?
>
> **Ans**: The prevailing belief is that early layers serve as the locations to encode lower-level information and for local information aggregation [1,2]. We believe at these layers the models are still **gathering local information** at each token position, leading to a greater flow from descriptions since they are closer to the answer position.
>
> [1] DoLa: Decoding by Contrasting Layers Improves Factuality in Large Language Models
>
> [2] Label Words are Anchors: An Information Flow Perspective for Understanding In-Context Learning
>
> [3] Locating and Editing Factual Associations in GPT
>
> ---
>
> > Q5: I think you don’t need to introduce the reversal curse again in related work.
>
> **Ans**: Thank you for your suggestion. We aim to emphasize the current research status on the reversal curse in this section. We will condense this part after revision.
>
> ---
>
> > Q6: Line 283: Can you say more specifically how this outcome diverges? What did [52] find?
>
> **Ans**: In [52], it was demonstrated that the inclusion of exemplary QAs during the training process improved models' performances on test QAs, and it was concluded that these QA examples enhanced the models' knowledge application abilities. However, we did not observe the same improvement in our experimental settings. We attribute this inconsistency to the impact of the thinking bias on the DescriptionIsName group.
>
> ---
>
> > Q7: I think you should discuss Limitations and Future Work in the main text.
>
> **Ans**: We will incorporate the Limitations and Future Work section back into the main text after revision.
>
> ---
>
> > L1: Maybe one thing to mention is that as far as I can tell, they only study one specific dataset.
>
> **Ans**: In fact, we have extended our experimental settings to **a new literature dataset, Book-Story**, to explore the potential broader implications of this bias across different types of data. We briefly mention this experiment in our main text (lines 133-135) and place the detailed description in Appendix D.

---

> ### Comment · Reviewer_p2dN · 2024-08-11
>
> Thanks for your detailed responses and clarifications (including the new results in the other rebuttals)! I have read them and have found them interesting and convincing. I will increase my score by one point.
>
> One comment regarding presentation: I understand that it's useful to refer to the reversal curse in this paper (including in title, abstract, introduction). I still think that your main contributions (thinking bias in multiple-choice Q&A) are separate from the reversal curse and study a different (but very related) phenomenon.

---

> > ### Author Response · Authors · 2024-08-12
> > **Appreciation to Reviewer's Response**
> >
> > Dear Reviewer p2dN,
> >
> > Thank you for dedicating your time to read our rebuttals and for raising your score! We sincerely appreciate your constructive comments for enhancing the presentation of our paper. These comments will be carefully considered when introducing the contributions of our work in the final version. We hope that all readers of our work will find their interest and inspiration in the experimental phenomena and our interpretations of today's LLMs' abilities.
> >
> > ---
> >
> > Best regards,
> >
> > Authors

---

### Official Review · Reviewer_PVmx · 2024-07-10

**Soundness:** 2
**Presentation:** 3
**Contribution:** 2
**Rating:** 7
**Confidence:** 4

**Summary:**

Building off prior work that studies the “reversal curse” in LLMs, the present paper provides additional analysis on 1) characterizing the limitations of LLMs on the reversal curse through more detailed experimentation (e.g., limitations with chain of thought prompting or providing multiple choice questions), and 2) interpreting the reasons as to why LLMs are biased towards correctly answering A is B when A is a name/proper noun. The main findings report that LLMs can improve generalization on B is A when 1) the prompt includes a multiple choice question; 2) LLMs are biased towards A (proper noun) is B (description), since this is how facts are typically represented in the training corpus; 3) This existing negative bias in LLMs cannot be mitigated by training/finetuning alone.

**Strengths:**

1)	The paper addresses a timely question
2)	The paper was, for the most part, straightforward and easy to understand.
3)	The experiments are interesting, and the effects appear strong

**Weaknesses:**

One of the main claims, that LLMs can disproportionately perform “NameIsDescription” correctly, is due to the fact pretraining datasets are biased towards having text in the form of NameIsDescription (i.e., A is B), but not the reverse. Despite the claim being mentioned several times, and though despite the claim being intuitive, the paper does not empirically quantify or demonstrate this that I could find. Are statistics reported of how often “A is B” is exhibited in the training documents relative to “B is A”? And is performance of the LLMs proportional to the ratio found in training documents? There seems to be some reference to Berglund et al., but this is a result of a prior paper, not the present paper. It would be helpful to quantify how biased LLMs are for A is B vs. B is A relative to the proportion they are exhibited in training data.

I found figure 3 to be confusing. What are the different colors supposed to indicate? (There’s no associated color bar.) Also, the incorrect answer of D makes it appear as if the incorrect selection was due to the tokenization of the name Graham Redwood.

There is also an issue of novelty – many of the reported results do not seem to particularly ‘novel’, perhaps because the results seem almost obvious. I think it would significantly help if the authors were to more clearly delineate their work from prior work in the Introduction, and to “signpost” exactly what the specific contributions of this work are (relative to prior work).

**Questions:**

1. Is the finding that names are easier to trigger recall potentially due the fact that names typically have fewer number of tokens than descriptions? Prior work has shown that the token-wise MLP layer of the transformers act as key-value memories (Geva et al. 2021). If there are fewer tokens associated with a name, wouldn’t it be easier to coordinate retrieval of memories (i.e., facts) across fewer token streams, rather than to coordinate memory retrieval across the many streams that comprise of “Description” tokens?

2. Related to a weakness mentioned above: What are the statistics/proportions that show a bias in the pretraining corpus of “A is B” over “B is A”? And how does that proportion match with actual LLM performance?

3. In multiple choice question prompting, are LLMs biased towards any particular response (e.g., ‘A’, ‘B’, ‘C’, or ‘D’)? I’m curious to know if the attention weights in a decoder-only model could potentially bias the model to retrieve more facts associated with ‘D’, since it is later in the prompt.

4. I’m skeptical of the interpretation (or over-interpretation) of “information flow” by computing the average attention weights to a given token. This concern is compounded by the fact that the models they used (Llama) are decoder-only models, which, by construction, have greater attention weights towards tokens presented later in the prompt. Might this metric be confounded by this (results in Fig. 2 & 3)?

Geva, Mor, Roei Schuster, Jonathan Berant, and Omer Levy. “Transformer Feed-Forward Layers Are Key-Value Memories.” In Proceedings of the 2021 Conference on Empirical Methods in Natural Language Processing, 5484–95. Online and Punta Cana, Dominican Republic: Association for Computational Linguistics, 2021. https://doi.org/10.18653/v1/2021.emnlp-main.446.

**Limitations:**

The authors claim that the curse of reversal can be somewhat mitigated if multiple choice questions are used. However, this appears to be a major limitation, while also being a strange suggestion – incorporating multiple choice questions assumes that the prompter knows the correct answer. Thus, in what scenario would this be helpful, aside from evaluating and adjudicating performances of multiple models?

---

> ### Author Rebuttal · Authors · 2024-08-07
>
> Thank you for your constructive suggestions and thoughtful comments. We hope that our response will effectively address your concerns.
>
> > W1 & Q2: What are the statistics/proportions that show a bias in the pretraining corpus of "A is B" over "B is A"?
>
> **Ans**: We conduct a statistical analysis on the English Wikipedia corpus [1]. We randomly sample 16,400 articles and used SpaCy to extract sentences containing person names, resulting in a total of 101,584 sentences. We then employ LLaMA3-70B-Instruct to judge whether the given sentence is: (1) a valid sentence and (2) uses a person's name as its subject. The results indicate that **76.9%** of valid sentences meet the criterion. Based on this and our original results, we believe there is a strong causal link between the data bias and the existence of the thinking bias. We leave a strict quantification of this bias to LLMs' performances for future works.
>
> [1] https://huggingface.co/datasets/wikimedia/wikipedia
>
> ---
>
> > W2: I found figure 3 to be confusing.
>
> **Ans**: Apologies for the confusion. The colors in the figure represent the saliency scores, ranging from low (green) to high (red). And the incorrect answer 'D' and the tokenization of 'Graham' is simply a coincidence in our test examples. We will add a legend to explain the colors and include more case studies after revision.
>
> ---
>
> > W3: There is also an issue of novelty – many of the reported results do not seem to particularly 'novel'.
>
> **Ans**: Thank you for your suggestion. We have provided a detailed discussion of the novelty and contributions of our work in the Global Rebuttal. We hope this will address your concern.
>
> ---
>
> > Q1: Is the finding that names are easier to trigger recall potentially due the fact that names typically have fewer number of tokens than descriptions?
>
> **Ans**: To study whether the number of tokens affects the efficiency of LLMs' memory retrieval, we conduct a new experiment using data with extremely long names, such as "Archibald Wolfgang Montgomery Beauregard Fitzwilliam the Third". We replace each name in the original dataset with these names, resulting in 2 new sets: LongNameIsDesc and DescIsLongName. The average number of tokens of new names and descriptions is **21.8** and **19.2**, respectively. We reconduct our main experiment and report results in Table 2 in the PDF. Given the performance on MCQs for LongNameIsDesc still significantly exceeds that of DescIsLongName, we conjecture that the models are still biased towards these long names under the effect of thinking bias.
>
> ---
> > Q3: In multiple choice question prompting, are LLMs biased towards any particular response (e.g., ‘A’, ‘B’, ‘C’, or ‘D’)?
>
> **Ans**: To study whether LLMs are biased towards certain options in fact retrieval, we model MCQs as a 4-label classification problem and calculate the models' performances on the NameIsDescription group. The DescriptionIsName group is excluded because the models' random selection behaviors on this set make it uncertain to derive meaningful interpretations. The results are posted in Table 3 in PDF. The variations in F1-scores across options fall within $\pm 5\%$, indicating **no strong preference** towards certain options in fact retrieval. Some fluctuations have been observed in the distribution of precisions and recalls. We attribute this to previous observations [1,2] on the model's inclination towards specific options when they are uncertain about the answer (higher recall always accompanies lower precision).
>
> [1] Beyond Performance: Quantifying and Mitigating Label Bias in LLMs
>
> [2] Language Models (Mostly) Know What They Know
>
> ---
>
> > Q4: I’m skeptical of the interpretation (or over-interpretation) of "information flow" by computing the average attention weights to a given token.
>
> **Ans**: We respectfully disagree with this view. As indicated in our earlier response, we did not observe a strong correlation between the options and the behavior of models. Additionally, as demonstrated by a counterexample in Fig. 2, in N2D MCQs where the descriptions are presented later in the prompt, the descriptions only exhibit relatively high saliency scores in the early layers. At the middle and later layers which are considered crucial for fact retrieval and semantic processing [1,2], the models still allocate a disproportional amount of information flow towards the name, despite the greater text distance.
>
> [1] Locating and Editing Factual Associations in GPT
>
> [2] DoLa: Decoding by Contrasting Layers Improves Factuality in Large Language Models
>
> ---
>
> > L1: The authors claim that the curse of reversal can be somewhat mitigated if multiple choice questions are used. However, this appears to be a major limitation.
>
> **Ans**: The primary goal of our work is not to directly mitigate the reversal curse, but rather to **examine and challenge** the widespread belief about this curse and LLMs' generalization abilities in more evaluation tasks which include MCQs. We do observe that when provided with the appropriate form of training documents and the presence of both A and B, LLMs can overcome the reversal curse. However, we interpret this as strong evidence that underscores the importance of the appropriate structure of factual knowledge for knowledge injection and downstream performance.
>
> Furthermore, although not the primary focus of our study, the observed mitigation effect in MCQs may have broader applications. For example, in RAG system, the retriever may retrieve multiple documents containing both correct and incorrect answers. Therefore, the ability to identify the correct answer would significantly impact the overall efficiency. In CoT settings, these contextual hints could also be produced by LLM itself, if the user could first help to narrow down the search range and then ask the LLM to list a few possible candidate answers before answering. Overall, we believe our new findings could contribute to both LLM interpretation and their application in future works.

---

> > ### Comment · Reviewer_PVmx · 2024-08-09
> > **Reviewer response to author rebuttal**
> >
> > I thank the authors for thoroughly engaging with my questions, and am impressed with the amount of work they have been able to perform during the rebuttal period. I think this paper would be a great contribution to NeurIPS, and commend the authors on their study. I will increase my score to 7.

---

> > > ### Author Response · Authors · 2024-08-10
> > > **Appreciation to Reviewer's Response**
> > >
> > > Dear Reviewer PVmx,
> > >
> > > Thank you for your timely feedback. We sincerely appreciate your time in reassessing our work and rebuttals and are grateful for your recommendation of our work to NeurIPS. We have and will continue to improve our work based on your valuable comments as well as those from other reviewers.
> > >
> > > ---
> > > Best regard,
> > >
> > > Authors

---

### Official Review · Reviewer_Mp9z · 2024-07-14

**Soundness:** 3
**Presentation:** 3
**Contribution:** 2
**Rating:** 5
**Confidence:** 3

**Summary:**

The authors extend the original reversal curse dataset to two tasks: open ended QA and MCQ. The authors analyze the generalization capabilities of LLMs on reversal tasks and provide several experiments towards their claims. They show that LLMs can generalize from A is B training to B to A, when both B and A are present in the question. Some of the results are expected based on existing results. For me the most significant contribution was the result in the relative intensities across layers. I may be willing to increase my score based on responses to certain questions.

**Strengths:**

- Strong applied work. The experiments are well posed (if only for small models), and do support the papers claims.
- In a welcome change from literature, the authors also provide negative results of their experiments in Section 4. While this is great, it is not clear from the paper how to mitigate the reversal curse on the said datasets.
- I liked the saliency score used in Section 3.2. Specifically, using saliency as an importance score for the l'th layer was insightful. I was wondering if this could be written as a dot product instead of a Hadamard product?

**Weaknesses:**

- Is the paper analyzing reversal curse? Was that not already done by the original reversal curse paper?
- The authors claim that generalization occurs when both A and B are present in the question. Is this possibly due to the structure of attention?
- Figure 5 shows that the models have a decreased accuracy when CoT is used. Why so? Should CoT not improve the results (as per earlier descriptions in the paper)? Any discussion on this?

**Questions:**

- Line 115 Was the model overfit to the fine-tuning data?
- What is the authors explanation for MCQs on NameIsDescription exhibiting generalization whereas DescriptionIsName not exhibiting the same?
- Does thinking step by step (CoT) put the name is description into context? Since in-context does not have the reversal curse problem, does CoT fix reversal curse on the given datasets?
- Is there an equivalent Table 4 identifying whether the Description is present in the CoT? Perhaps both Name as well as Description can exist in the CoT?
- Figure 5 goes against existing literature [52] as noted by the authors themselves. "We attribute this divergence to the structure of the training documents since their training samples mainly use names or personal pronouns as subjects, which generally mirrors the structure of the NameIsDescription subset." The description provided was not clear to me.
- Why not use the saliency scores of particular tokens, like the names, instead of a particular position?
- Snt and Sdt are great observations. Can these be used to actually modify the attention? i.e. since we have white-box access to the model, can one actually increase the attention manually to check if this (a) solves the RC problem, or (b) improves generalization in terms of accuracy?
- Can the authors clearly outline why their contribution is novel, and what are the major take-aways from their work?

### Suggestions:

- Based on the introduction, I could not understand the paper contributions. Can the paper provide a short section or paragraph on that in Section 1?
- The legend in Figure 5 is unclear. Does it mean that OOD are the smaller bars which are having the masks?

**Limitations:**

- No major limitations. I feel some of these results are expected based on Berglund et al.
- My concern is on novelty. I am hoping the authors can address this question in the rebuttal.

---

> ### Author Rebuttal · Authors · 2024-08-07
>
> # Part 1
>
> > W1: Is the paper analyzing reversal curse? Was that not already done?
>
> **A**: Yes, we thoroughly analyze the reversal curse. As discussed in Global Rebuttal, we find previous discussions on this problem are **rather vague and arbitrary**. We provide conceptual clarity and an explanation of the original reversal curse (Reviewer e5Vg), which has not been done by previous works. A detailed disucssion is given in our Global Rebuttal.
>
> ---
>
> > W2: Is the generalization that occurs when both A and B are present in the question due to the structure of attention?
>
> **A**: Yes, but it is only a necessary condition. Another important factor is the structure of training documents, which must be organized to align with LLMs' thinking bias, such as NameIsDescription. The underlying reason we believe is that LLMs' pretraining corpus exhibits a bias towards these structures, as supported by our response to Reviewer PVmx's W1. This distribution bias shapes LLMs' recall process from Name to Description, but not vice versa.
>
> ---
>
> > W3: Figure 5 shows that the models have a decreased accuracy when CoT is used. Why so?
>
> **A**: The reason for the decreased accuracy when CoT is used in training is that the test is conducted to force models to output **without CoT steps**. This setting is intended to explore whether models can learn the reasoning paths from the CoT QA data and enhance the performance even when CoT is not utilized in the test [1,2]. The results when CoT is used in test are reported in Table 1 from the PDF, with no decrease observed.
>
> [1] Large language models can self-improve
>
> [2] Let’s think dot by dot: Hidden computation in transformer language models.
>
> ---
>
> > Q1: Line 115 Was the model overfit to the fine-tuning data?
>
> **A**: We disagree. To avoid overfitting, we use the hyperparameter shown in Table A5 and data augmentations for training. The performance of our models on MMLU in Table A6 also suggests that no obvious overfitting phenomenon occurs. Furthermore, we re-run the training of LLaMA2-7B and 13B chat and report the training and testing curves in Figure 1 of our PDF. Again, no overfitting was observed.
>
> ---
>
> > Q2: What is the explanation for MCQs on NameIsDescription exhibiting generalization whereas DescriptionIsName not exhibiting the same?
>
> **A**: The direct cause is that the models exhibit a bias towards recalling information related to names instead of descriptions when both are presented in the question. For DescriptionIsName, all models fail to recall the correct descriptions given the paired names, which has been observed by our Open-QA experiments. The root cause, we believe, is that the pretraining corpus of LLMs is more biased towards the expression of NameIsDescription, which potentially shapes the preference of the above thinking patterns in LLMs.
>
> ---
>
> > Q3: Does thinking step by step (CoT) put the name is description into context? Since in-context does not have the reversal curse problem, does CoT fix reversal curse on the given datasets?
>
> **A**: No. The CoT prompt we use is shown in section 3.1, line 152. Rather than directly providing the related training documents as context, we encourage LLMs to **generate this information themselves** based on their knowledge. This approach also gives us an understanding of their problem-solving process.
>
> To your second question, we re-ran the MCQ tests and included ground-truth information related to all options in the context of the input query. The average MCQ accuracies across both NameIsDescription and DescriptionIsName on LLaMA2-7B and 13B are 96.4 and 98.2. The results suggest that it can be fixed if all ground-truth information is presented, but empirically can be hard to achieve in real applications.
>
> ---
>
> > Q4: Is there an equivalent Table 4 identifying whether the Description is present in the CoT?
>
> **A**: In our CoT experiment in sec. 3.1, we observe that **both names and descriptions** exist in the models' self-generated CoT steps. It is their recalling behavior (i.e., whether their recalling starts with names or descriptions) that we are more interested in. Therefore, in Table 3, we count the frequencies of the pattern "[Name] is [Description]" observed in their CoT steps, such as "Daphne Barrington is the director of 'A Journey Through Time'", but not "The director of 'A Journey Through Time' is Daphne Barrington".
>
> ---
>
> > Q5: The description discussing the divergence between Figure 5 and [52] is not clear.
>
> **A**: In [52], the authors show that the inclusion of exemplary QAs during training could enhance models' performances on test and conclude that these QA examples enhance models' knowledge application abilities. However, we did not observe the same improvement in our experiment. We attribute this inconsistency to the impact of the thinking bias on the knowledge application abilities within the DescriptionIsName group.
>
> ---
>
> > Q6: Why not use the saliency scores of particular tokens, like the names, instead of a particular position?
>
> **A**: The positions used for computing saliency scores are exactly those of the name tokens.
>
> ---
>
> > Q7: Snt and Sdt are great observations. Can these be used to actually modify the attention?
>
> **A**: Yes. Due to time limitations, we only experimented with heuristic methods on D2N tasks of MCQs from the DescriptionIsName group. To force models to utilize more information from descriptions, we amplify the attention score of descriptions by a factor of 2 and decrease that to the names by a factor of 0.2. This operation is applied to all attention heads from layer 10 to 30 of our trained LLaMA2-7B-chat model. The resulting correctness is boosted to 44.2%. But we also notice some strange behaviors after our brutal editing, including the output of mismatched option labels and textual content. We believe that a more intricate editing method, such as circuit finding [1], could address this issue. We leave this for future work.
>
> [1] Localizing Model Behavior with Path Patching

---

> ### Author Response · Authors · 2024-08-07
> **Rebuttal-Part 2**
>
> Dear Reviewer Mp9z:
>
> We are sorry that we were unable to respond to all your questions in a single rebuttal section due to the character limit. We have included our responses to the remaining questions here for your reference, as well as for the benefit of other reviewers. We apologize for any inconvenience this may have caused.
>
>
> # Part 2
>
> > Q8: Can the authors clearly outline why their contribution is novel, and what are the major take-aways from their work?
>
> **A**: We hope our response in Global Rebuttal will effectively address your concerns.
>
> ---
>
>
> > Suggestion: In Figure 5, does OOD are the smaller bars which are having the masks?
>
> **A**: Yes. And we will improve the clarity of this legend in the revised version of the paper.
>
> ---
>
> > L1 & 2: I feel some of these results are expected based on Berglund et al. My concern is on novelty.
>
> **A**: We respectfully disagree. We believe that the only experimental results that can be expected based on RC are the model's performance on Open-QA. The generalization difference between the NameIsDescription and DescriptionIsName groups, particularly the phenomenon that the models cannot identify the correct answer even though they can directly answer the original question without options, is **completely unexpected and even counter-intuitive**. It goes against the common belief that identifying the correct answer would be much easier than producing it from scratch. We hope that the novelty listed in the Global Rebuttal could adequately address your concerns.
>
> ---
>
> > Additional Question: Can dot product replace Hadamard product in saliency score computation?
>
> **Ans**: The incorporation of Hadamard product in Equation 1 is inspired by one of the first works [1] that introduced saliency analysis into language models. Suppose a new mask variable $Z\in \mathbb{R}^{L\times L}, Z\in [0, 1]$ is applied to the attention matrix $A \in \mathbb{R}^{L\times L}$ to control the interaction between token $i$ and token $j$, which gives us $A'=A \odot Z$. The sensitivity of the model to the mask variable $Z$, $I_{ij} = |\frac{\partial \mathcal{L}}{\partial Z_{ij}}|$ can be seen as the importance score of the interaction between token $i$ and $j$. After applying the chain rules, the final expression for $I$ as well as our definition of the saliency score can be written as $I_{ij} = |A \odot \frac{\partial \mathcal{L}}{\partial A}|_{ij}$. We are not sure whether dot product could serve the same purpose as well. But we remain open to any further discussion.

---

> > ### Comment · Reviewer_Mp9z · 2024-08-13
> >
> > Thanks to the authors. It addressed some questions I had during my initial reading of the work. I hope, if the paper gets accepted, the authors can move some of these points into the main body of the work for camera ready.
> >
> > I will be increasing my score, as the rebuttal addresses some of my earlier concerns. I have no objections to the paper getting accepted. There are no technical concerns, or issues related to the soundness of the paper.

---

> > > ### Author Response · Authors · 2024-08-13
> > > **Appreciation to Reviewer's Response**
> > >
> > > Dear Reviewer Mp9z,
> > >
> > > Thank you for your feedback! We are deeply grateful for your time in reading our rebuttals and your willingness to reassess our work. In response, we will further polish our final paper based on the valuable points raised by you and other reviewers.
> > >
> > > ---
> > >
> > > Best regards,
> > >
> > > Authors

---

### Author Rebuttal · Authors · 2024-08-07

We thank all the reviewers for their time, insightful suggestions, and valuable comments. We also notice that there may be concerns regarding the comparison of our work to previous studies on the reversal curse and the clarity of our contributions. Here, we provide a comprehensive summary of the reviewers' feedback and outline the novelty of our work, as well as our contributions and main takeaways.

# Merits of our work

The reviewers have acknowledged the strengths of our papers as follows:

* Our findings on LLMs' generalization abilities are timely, interesting, and novel. (Reviewer PVmx, Reviewer p2dN, Reviewer e5Vg)
* The experiments are extensive, insightful, and compelling in supporting the papers' claims. (Reviewer Mp9z, Reviewer PVmx, Reviewer p2dN, Reviewer e5Vg)
* Our writing is straightforward and easy to understand. (Reviewer PVmx, Reviewer p2dN)

# Novelty of our work comparing to previous studies

The reversal curse [1] is proposed based on the observation that models trained on "A is B" cannot complete the sentence "B is ...". Before this work, there is no convincing explanation or even a clear definition of this curse. For example, the original paper [1] suggests that it is *"a failure of basic logical deductions from the training documents"*, which is rather vague and acknowledged as impossible to verify. A later work [2] claims that *LLMs trained on "A is B" cannot answer questions related B as long as the relationship "A is B" is not provided by the context.* These claims raise concerns about the generalization ability of today's LLMs: *do LLMs understand their training documents, such as the equivalence between A and B? If they do, to what extent can they apply this knowledge to downstream tasks?*

Our work takes a **further step** towards answering the above question as well as the examination of these previous claims regarding the RC problem. To be more specific:
1. We **refute** the previous belief that LLMs can only realize or answer "B is A" when "A is B" is given in the context, as evidenced by the success of MCQs from the NameIsDescription group. This finding suggests that it is unfair to simply conclude RC as an inability to comprehend the training documents. It is more likely to be a backward recall deficiency (Reviewer e5Vg).
2. LLMs' abilities to apply their training knowledge can strongly correlate with the structure of the training documents. An interesting example is that, even when the models are able to **answer the question directly**, their ability to identify the correct answer from options could be **no better than random guessing** (i.e. Open-QA D2N V.S. MCQ D2N on the DescriptionIsName set). This gives us an unexpected, even counter-intuitive observation on LLMs' abilities.
3. We discover that LLMs display a bias toward using names to initiate their analysis of the query and the retrieval of related information, which explains the observed phenomenon and is **rigorously evidenced** by our comprehensive interpretation experiments (all reviewers). This finding underscores the importance of the structure of training documents on LLMs' downstream performances. We believe our work serves as the corner stone towards the mitigation of this generalization deficiency and provide varible insights for the development of more effective knowledge injection techniques.

# Contributions & Main Takeaways

* **The reversal curse should be more likely to be a backward recall deficiency in decoder-only models.** The success on the MCQs serves as a counterexample to the previous claim that LLMs cannot understand the equivalence between "A is B" and "B is A" in their training documents.
* **Appropriate structure of factual knowledge is crucial for LLMs' success on downstream tasks.** Training data adhering to specific structures, such as NameIsDescription or Book-Story, enables models to provide correct answers when sufficient leads (i.e., available options) are provided. However, when training documents deviate from the models' preferred forms, their knowledge application behaviors become unstable and even counter-intuitive (i.e., Open-QA D2N V.S. MCQ D2N).
* **LLMs display a bias toward using names to initiate their analysis of the query and the retrieval of related information.** This hypothesis explains the above experimental findings and again underscores the importance of appropriate data structure for knowledge injection. Furthermore, this finding also raises a series of new questions beyond the discussion of the original RC problems: when and what shapes such a thinking pattern? How to mitigate its negative effect? Could other LLMs' deficiencies such as hallucination [3] or social bias [4] be related to it? All these questions remain yet to be explored and would enhance our understanding of today's LLMs.

[1] The Reversal Curse: LLMs trained on "A is B" fail to learn "B is A"

[2] Are We Falling in a Middle-Intelligence Trap? An Analysis and Mitigation of the Reversal Curse

[3] Siren's Song in the AI Ocean: A Survey on Hallucination in Large Language Models

[4] Bias and Fairness in Large Language Models: A Survey

---

### Decision · Program_Chairs · 2024-09-25

**Decision:**

Accept (poster)

**Comment:**

The reviewers and I generally liked this paper, finding it to be relevant, timely, and interesting.  The paper is well-written and provides actionable insights into the inner workings of LLMs, which is a topic of perpetual interest. The authors have also done a good job of responding to reviewers suggestions, and I have confidence that the final version of this paper will be even better than this version.